



# Copper accelerates photochemically induced radical chemistry of iron-containing SOA

Kevin Kilchhofer[1,2], Markus Ammann[1], Laura Torrent[1,3,4], Rico K. Y. Cheung[1], and Peter A. Alpert[1,5]

[1]PSI Center for Energy and Environmental Sciences, 5232 PSI Villigen, Switzerland
[2]Department of Environmental System Science, Institute for Atmospheric and Climate Science, ETH Zurich, 8092 Zurich, Switzerland
[3]School of Architecture, Civil, and Environmental Engineering (ENAC IIE GR-LUD), École Polytechnique Fédérale de Lausanne (EPFL), 1015 Lausanne, Switzerland
[4]now at: Department of Chemistry, Faculty of Sciences, University of Girona, 17003 Girona, Spain
[5]now at: XRnanotech, 5232 Villigen PSI, Switzerland

**Correspondence:** Kevin Kilchhofer (kevin.kilchhofer@psi.ch) and Markus Ammann (markus.ammann@psi.ch)

**Abstract.** Photochemical aging in secondary organic aerosol (SOA) particles alters their chemical composition and affects their adverse health effects. However, there is limited mechanistic insight on the role of transition metals in photochemical SOA aging and the evolution of the oxidative potential through their effect on radical chemistry. Here, we investigated the influence of copper (Cu) on the photochemical aging of iron (Fe) containing SOA in single particles using scanning transmis-
5 sion X-ray microscope measurements and chemical box modeling. The SOA proxy included citric acid (CA), iron(III) citrate ($Fe^{III}(Cit)$) and copper(II) citrate ($Cu^{II}(HCit)$), which were exposed to UV light ($\lambda = 365\,\mathrm{nm}$) in a humidified environmental cell. We modeled known catalytic radical destruction mechanisms resulting from cross-redox reactions between copper and iron. Simulating anoxic $Fe^{III}(Cit)/Cu^{II}(HCit)/CA$ aging experiments showed a lower initial iron(III) reduction compared to $Fe^{III}(Cit)/CA$ particles, indicating a reduced iron(II) quantum yield than the photolysis of the $Fe^{III}(Cit)$ alone. We hypothesize
10 that this effect may be due to copper replacing an iron center in a polynuclear complex. At higher relative humidity (RH) up to 60%, a lower iron(II) quantum yield could not account for our observations of iron reoxidation in the dark. Instead, reoxidation appears to be highly sensitive to a potential copper(II)-induced reoxidation reaction. We provide a comprehensive discussion and evaluation of the poorly understood role of copper in modifying redox and radical chemistry, which is relevant for reactions involving transition metals mixed with SOA in the atmosphere.



## 1 Introduction

Secondary organic aerosol (SOA) particles are a major component of tropospheric particulate matter (PM), predominantly formed through the oxidation of biogenic and anthropogenic volatile organic compounds (Hallquist et al., 2009). Consequently, SOA particles consist of various molecules from different precursor gases, which greatly influence atmospheric chemistry (Pöschl and Shiraiwa, 2015). Due to their substantial mass yield, SOA particles are significant contributors to particulate air pollution (Huang et al., 2015). The health impacts of PM are suggested to be related to the concentration of reactive oxygen species (ROS), such as hydroxy radicals or hydroperoxides (Weichenthal et al., 2019; Shiraiwa et al., 2023). ROS can be introduced into the body through inhalation of PM (particle-bound ROS) (Dellinger et al., 2001) or catalytically generated in vivo after inhalation by redox-active PM species (semi-empirically quantified by oxidative potential, or OP) (Bates et al., 2019). For example, particle-bound ROS have been associated with fine-mode SOA particles and with coarse-mode metals from vehicular non-exhaust emissions (Daellenbach et al., 2020). SOA particles nucleated from the oxidation products of biogenic volatile organic compounds (VOC) can mix with primary emitted aerosol particles (e.g. mineral dust), which contain redox-active components such as transition metal ions (Charrier and Anastasio, 2015). This mixing has been found to enhance ROS production by these particles (Tuet et al., 2017; Alpert et al., 2021).

Transition metal ions, such as iron and copper, have long been known to dissolve in atmospheric liquid phases (Kieber et al., 2001; Deguillaume et al., 2005). Natural emissions from dust regions and anthropogenic activities, such as traffic and combustion, are primary sources of soluble iron (Ito and Miyakawa, 2023), making it the most abundant metal in the atmosphere. Copper emissions increased greatly during the industrial revolution (Hong et al., 1996), and the atmospheric copper concentration have been measured up to one-tenth of ambient iron concentration (Schroeder et al., 1987). In megacities, higher metal concentrations lead to increased levels of soluble iron and copper in PM (Chifflet et al., 2024). In SOA, dissolved metals form complexes with organic material (Moffet et al., 2012; Al-Abadleh, 2015, 2024). Field studies have shown that soluble iron occurs mainly in complexes with carboxylate functional groups (Tapparo et al., 2020; Tao and Murphy, 2019) and both iron and copper complexes have been identified in ambient PM, indicating a significant alteration of their OP (Wei et al., 2019).

Due to the long atmospheric lifetimes, aging processes can significantly alter the physicochemical properties of SOA particles (Kroll and Seinfeld, 2008; Robinson et al., 2007). Photochemistry plays a central role in many aerosol aging processes (George et al., 2015). In the condensed phase, photochemical reactions are more easily initiated compared to the gas phase, as photoexcited species are in closer proximity to energy or electron acceptor. Organometallic complexes often have a high UV absorption cross section, leading to indirect photochemistry (George et al., 2015; Corral Arroyo et al., 2018). We hypothesize that particles containing iron, copper, and SOA undergo photochemical reactions subsequently producing ROS. This study aims to quantify the redox chemistry of these transition metals and further elucidate the photochemical reaction mechanism, shedding light on new pathways of ROS formation in SOA.

Citric acid (CA) is commonly used as a model system for SOA due to its oxygen to carbon ratio (O:C) and atmospherically relevant viscosity properties (Reid et al., 2018; Dou et al., 2021; Alpert et al., 2021; Wang et al., 2023). Iron citrate ($Fe^{III}Cit$) is a representative iron-carboxylate complex with reasonably well-characterized photochemistry (Cieśla et al., 2004; Wang





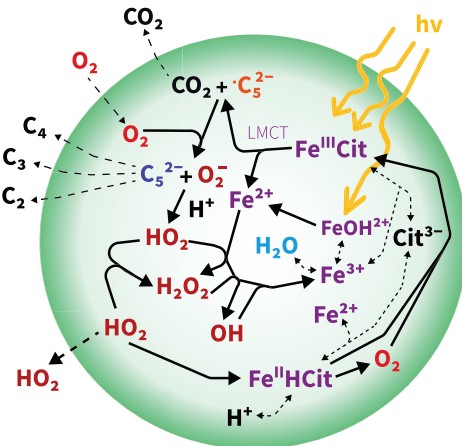

**Figure 1.** Schematic of photochemically induced $Fe^{III}(Cit)/CA$ chemistry in a viscous particle. The detailed mechanism, including equilibria and reaction rates, is outlined in Tables 1, 2 and 3. $^{\bullet}C_5^{2-}$ denotes the carbon-centered free radical $^{\bullet}C_5H_5O_5^{2-}$, while $C_5^{2-}$ refers to the ketoacid $C_5H_4O_5^{2-}$ in reactions E8, R1 and R2. $Fe^{III}Cit$ denotes $FeC_6H_5O_7$, $Fe^{II}HCit$ corresponds to $FeC_6H_6O_7^{2-}$, and $Cit^{3-}$ refers to $C_6H_5O_7^{3-}$. LMCT stands for ligand-to-metal charge transfer.

et al., 2012; Weller et al., 2013, 2014; Dou et al., 2021). Its high absorbance in the UV-visible range (Seraghni et al., 2012;
Abida et al., 2012) enables $Fe^{III}Cit$ to initiate condensed phase radical processes at wavelengths that most oxidized organic compounds (non-carbonyl) would not absorb on their own.

Speciation in $Fe^{III}Cit$ solutions is strongly influenced by metal redox reactions (Alpert et al., 2021). The number of citrate ligands per iron center, as well as the abundance of polynuclear (thus poly-iron) complex aggregates varies depending on pH and the ratio between CA and $Fe^{III}$(Abrahamson et al., 1994; Silva et al., 2009; Gracheva et al., 2022), with each species
exhibiting different absorbance. $Fe^{III}Cit$ photochemistry is initiated by ligand-to-metal charge transfer (LMCT), generating a reduced radical complex as the first intermediate. This may decompose into $Fe^{II}$ and a citrate radical, which rapidly decarboxylates at the central carboxyl group, releasing $CO_2$ and forming a carbon-centered free radical ($^{\bullet}C_5^{2-}$) (Weller et al., 2013). In the presence of $O_2$, ROS species, $HO_2$ and $O_2^-$, and a ketoacid ($C_5^{2-}$) are produced. The ketoacid can also form through the reaction of the citrate radical or the reduced radical complex with $Fe^{III}Cit$, or by reaction with another $Fe^{III}$ center within
the same polynuclear complex (Abrahamson et al., 1994; Glebov et al., 2011). These additional pathways yield a second $Fe^{II}$ without producing ROS, potentially leading to a higher iron(II) quantum yield ($\phi$). Iron(II) can be reoxidized to iron(III) with ROS via the Fenton reaction cycle (Fenton, 1894), and subsequently combines with another CA molecule, completing the photolytic cycle. This photodegradation process also is an important sink of carboxylate groups, and prolonged irradiation may result in the formation of Fe-carbonaceous colloids, as recently reported by West et al. (2023).
The formation of various ROS species, primarily driven by the generation of peroxy radicals, relies on $O_2$ supplied from the gas phase. Since $O_2$ has relatively low solubility, its availability can become rate limiting, especially when SOA particles



become highly viscous at low relative humidity (RH) and/or temperatures (Virtanen et al., 2010; Koop et al., 2011). Under these anoxic conditions within the particle bulk, highly reactive species generated by photochemistry can accumulate. Upon humidification to higher RH, the increase $O_2$ uptake enhances the formation of additional ROS (Alpert et al., 2021).

This work aims to bridge the gap between model SOA proxies and ambient SOA by investigating the role of relevant metal ions, specifically iron and copper, in the aerosol phase. The interactions between iron and other dissolved metals in SOA, and their impact on particle-bound ROS production, remains unexplored. Previous studies on this subject have primarily focused on the role of metals in surrogate lung fluid in relation to their OP (Charrier et al., 2014; Wei et al., 2019; Lelieveld et al., 2021, 2024). In this study, we introduce copper citrate ($Cu^{II}(HCit)$) to evaluate the influence of copper ions and organic com-

plexes on the photochemical aging of $Fe^{III}(Cit)$/CA particles, thereby examining its role in aerosol aging, multiphase chemistry, and particle-bound ROS formation. Using X-ray spectromicroscopy, we found that copper alters the photochemical aging process in $Fe^{III}(Cit)$/$Cu^{II}(HCit)$/CA particles, particularly in terms of iron reoxidation, compared to $Fe^{III}(Cit)$/CA particles studied by Alpert et al. (2021) without copper. This observation led us to model the photochemistry in $Fe^{III}(Cit)$/$Cu^{II}(HCit)$/CA particles and assess the importances of ROS-copper and iron-copper coupled reactions available from previous literature. Notably,

the new reoxidation experiments could only be explained by adjusting the iron(II) quantum yield of $Fe^{III}(Cit)$ photolysis. Thus, copper may not only catalyze ROS cycling but also potentially replace iron in polymetal complexes, reducing the photolysis quantum yield and accelerating reoxdiation in this system.



## 2 Material and Methods

### 2.1 STXM/NEXAFS for determining iron oxidation state.

Scanning transmission X-ray microscopy coupled with near-edge X-ray absorption fine structure (STXM/NEXAFS) spectroscopy (Raabe et al., 2008; Moffet et al., 2010) was used to determine the fraction of iron(III), $\beta$, out of the total iron after photoreduction and during in situ reoxidation in individual particles, following established methods (Alpert et al., 2019, 2021; Dou et al., 2021). The experiments were conducted at the PolLux beamline at the Swiss Light Source (SLS). NEXAFS spectra were also measured to quantify the oxidation states of both iron and copper. X-ray spectra were obtained by imaging particles

at individual X-ray energies and then scanning across the energy range of relevant absorption edges. For iron, the energy was scanned from 700 to 735 eV, covering the L-edges around 709 and 721 eV, while for copper, the scan ranged from 925 to 965 eV to capture the $L_{2,3}$-edge peaks at 934 and 954 eV. For carbon, the energy was scanned from 278 to 318 eV to cover the C-K-edge NEXAFS region.

At different oxidation states, iron has a different L-edge peak position, occurring at, e.g. $\sim$ 707.8 and 709.6 eV for $FeCl_2$

and $FeCl_3$, respectively, with a separation of 1.8 eV (Moffet et al., 2012). This information was used to calibrate X-ray energy (Figure A1) with a constant offset. A mixture of the two oxidation states was quantified by first subtracting the background absorption at the iron pre-edge (between 700-705 eV, where iron does not absorb strongly), then taking the ratio of the absorption peak heights, and applying a previously established parameterization (Moffet et al., 2012; Dou et al., 2021; Alpert et al., 2019, 2021). All particles appeared homogeneous in terms of metal and organic distribution within all particles, meaning

we never observed dense nanoparticles or gradients in Fe, Cu or organic concentration. However, we did observe gradients in oxidation state, and note that only Fe(II), Fe(III), Cu(I) and Cu(II) oxidation states were detected.

It enhances the time resolution when quantifying iron oxidation state, we followed the procedure by Alpert et al. (2019), calculating the pre-edge absorption from a polynomial fit as a function of the sum of the absorption at the Fe(II) and Fe(III) peaks. Then, two high spatial resolution images with a pixel size of $35 \times 35$ nm$^2$ were acquired at the peak absorption energies

for iron(II) and iron(III) at 710.1 and 711.9 eV, respectively (see Figure A1), allowing the calculation of $\beta$ in the particles using Equation 1. This fraction is then used to map the individual particles with a resolution of tens of nanometers, providing detailed spatial insights into their iron oxidation states.

$$\beta = \frac{C(Fe^{3+})}{C(Fe^{2+}) + C(Fe^{3+})} \tag{1}$$

When determining $\beta$, calculations were performed for each pixel in the images of particles, with uncertainties stemming

from photon counting statistics, nanometer scale drift in positioning, automated image alignment, and errors from the iron pre-edge calculation (prediction error at 95% confidence). Radial profiles of $\beta$ were obtained by averaging concentric pixels on and from the perimeter of the imaged particles (Alpert et al., 2019). The uncertainty in $\beta$ was determined to be either $\Delta\beta = \pm\, 0.07$, or propagated from the photon counting error, whichever was larger. Two experimental procedures were utilized to investigate photochemical reduction and reoxidation over time. In the first, particles were generated from a $Fe^{III}(Cit)/CuCl_2/CA$ solution,

placed under vacuum, and exposed to UV light. In the second, particles were generated from a solution containing $Cu^{II}(HCit)$





and placed in an environmental cell with controlled light exposure, RH, and $O_2$ concentration. When the humidity (and thus particle water content), $O_2$, or copper content was varied, the iron pre-edge absorption of the particles was different. Prior to light exposure for each experiment, background absorption at $700\,\mathrm{eV}$ and the two absorption peaks for iron(II) and iron(III) were measured for about 100 particles. This data was used to create a unique polynomial fit for background absorption prior to
UV exposure. As previously mentioned, the background absorption image was calculated from the fit, reducing the number of images from three to two in order to decrease the time needed to quantify $\beta$ maps (Alpert et al., 2021).

## 2.2  Solutions and sample preparation

Particles for STXM/NEXAFS analysis were prepared by nebulizing the solutions using a custom-built ultrasonic nebulizer (Steimer et al., 2015). The nebulized particles were then dried at RH $\leq 30\%$, and directed to a single jet impactor in a $N_2$ flow,
following passage through a differential mobility analyzer to select a mobility diameter. We prepared six different solutions. Three had a $1:1$ mole ratio of $Fe^{III}(Cit):CA$, two with mole ratios of $Cu^{II}(HCit):Fe^{III}(Cit) = 0.2$ and $0.02$, and one with a mole ratio of $Cu^{II}Cl_2:Fe^{III}(Cit) = 0.2$. These ratios were chosen to replicate $Fe^{III}(Cit)/CA$ experiments from Alpert et al. (2021) and to evaluate atmospherically relevant copper concentrations, approximately one-tenth of the iron concentration (Schroeder et al. (1987); Wei et al. (2019). CA ($\geq 99.5\%$; CAS = 5949-29-1), $Fe^{III}(Cit)$ tribasic monohydrate (18-20% Fe basis; CAS =
2338-05-8) and $Cu^{II}(HCit)$ (97%; CAS = 866-82-0) were purchased from Sigma-Aldrich. The dilute aqueous solutions were prepared in ultrapure water ($18\,\mathrm{M\Omega\,cm}^{-1}$, Milli-Q). The particles were impacted on silicon nitride windows, which were then sealed inside replaceable sample holders compatible with the environmental cell used for the STXM/NEXAFS (Alpert et al., 2021; Huthwelker et al., 2010).

## 2.3  STXM environmental cell and photochemical reactor

In the environmental cell, the iron oxidation state was mapped in situ in the dark following photolysis (Alpert et al., 2021; Huthwelker et al., 2010). The particle samples were first transported under vacuum to the PolLux endstation at the SLS, where they were loaded into the environmental cell. During preparation, transport and loading, all ambient room light was switched off, and efforts were made to shield the samples and the solutions from any additional light, ensuring minimal light exposure before sealing the end station. The interval between impaction and X-ray investigation was approximately two hours. The
environmental cell maintained a controlled gas flow with constant RH ranging from 0-60 % and $O_2$ at partial pressure adjusted according to the desired RH. Helium served as the carrier gas for $O_2$ and $H_2O$ with a total flow rate of $20\,\mathrm{mL\,min}^{-1}$ (stp). At RH = 60%, 50% and 40%, the oxygen to helium flow ratios were 0.6, 0.75 and 0.8, respectively.

Figure 2 shows a schematic of the UV-LED fiber optic system used for UV-aging experiments in the STXM. The system includes a LED (LuxiGenTM 365 nm UV LED Gen 4 Emitter, Model LZ1-00UV0R) and two interchangeable collimators
(Thorlabs, model CFC-2X-A and model CFC-5X-A), which provide uniform illumination throughout the sample area. The power density can be adjusted between $(2.36 \pm 0.06)\,\mathrm{W\,m}^{-2}$ and $(13.55 \pm 0.06)\,\mathrm{W\,m}^{-2}$) over a narrow wavelength range of 360-380 nm (with a full-width half-maximum of approximately $20\,\mathrm{nm}$). This corresponds to a photolysis frequency for $Fe^{III}(Cit)$, $j_{FeCit}$, ranging from $(1.4 \pm 0.2) \times 10^{-3}\,\mathrm{s}^{-1}$ to $(8.1 \pm 0.9) \times 10^{-3}\,\mathrm{s}^{-1}$. Light intensity was regulated by a dimmer



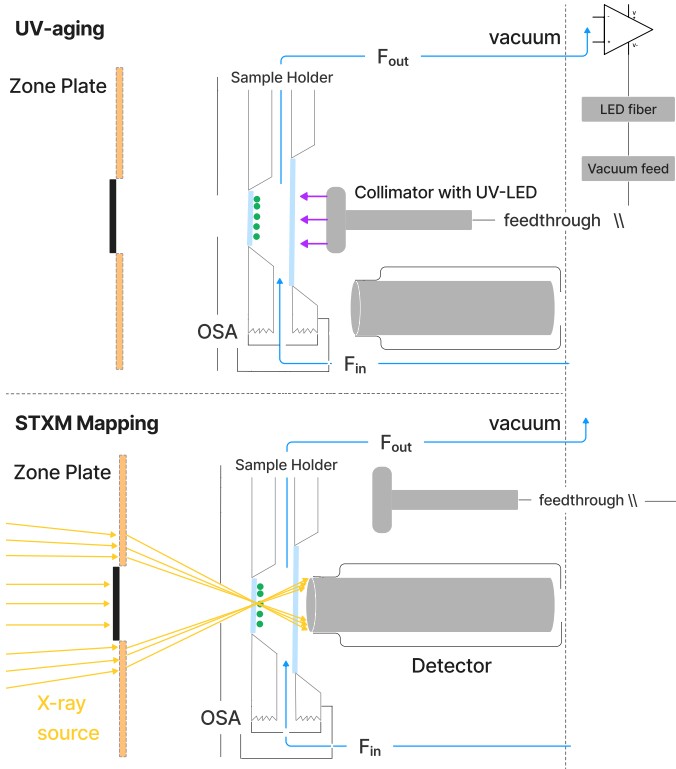

**Figure 2.** Schematic of the environmental cell situated inside the STXM/NEXAFS vacuum chamber. During UV-aging, the particles (green dots) on the silicon nitride (SiNit) membranes were aligned with a UV-LED collimator. For STXM/NEXAFS analysis, the collimator was moved aside and an X-ray detector was moved in its place. SiNit membranes were sealed in an aluminium sample holder, which was attached to the aluminium environmental cell sealed with an O-ring. A constant, humid helium flow (blue line) was maintained through the environmental cell. The positions of the detector and collimator were automatically alternated between the two experimental conditions.

circuit, which was controlled by an electrical signal ranging from 1 to 7 V. The fiber optics, with a numerical aperture of 0.5 and a core diameter of $1000\,\mu\text{m}$, allowed for a large acceptance angle, ensuring sufficient light transmission from the LEDs. The fiber is multimode and has a high OH content, which ensures transmission of ultraviolet and visible light with minimal loss. A custom-built vacuum feed through connected the fiber optics to the STXM chamber. The spectral flux density of the light as a function of wavelength was measured after the collimator using a pre-calibrated power meter (Air-Cooled Thermopile Sensors, Model PM2) and spectrometer (Avantes, model AvaSpec-ULS2048XL-EVOis).

## 2.4 Photolysis frequency

We compared the photolysis frequency of $\text{Fe}^{\text{III}}(\text{Cit})$, $j_{\text{FeCit}}$, when illuminated by the LED light, to the actinic flux in Los Angeles at sea level for June $(\text{E}_{\text{QF}}(\lambda)_{sun})$ and December days. The actinic flux data was obtained using the *libRadtran* software (Emde et al., 2016), with the following parameters, albedo = 0.1, aerosol optical depth, $\tau = 0.1$, single scattering albedo (SSA) =





0.85, solar zenith angle (SZA) = 23.4° and $O_3$ column = 300 DU. The corresponding spectra can be seen in Figure B1) in the
SI. At noon in June, the photolysis frequency in Los Angeles is $j = (2.9 \pm 0.2) \times 10^{-2} \, s^{-1}$, integrated over the 270-500 nm
wavelength range. The UV-aging of $Fe^{III}(Cit):Cu^{II}(HCit):CA$ particles under atmospheric conditions scales in proportion to $j$
calculated from the LED flux, the absorption cross section, and an iron(II) quantum yield, $\phi$, of 1. For the wavelength range
emitted by the LED (350-400 nm), $j_{LED}$ reaches $(8.1 \pm 0.1) \times 10^{-3} \, s^{-1}$.

Using an equation derived by Malecha and Nizkorodov (2016) (Eq. 2) we compared $j$ under atmospheric conditions, with
the LED spectrum for two wavelength ranges, 350-400 nm and 270-500 nm. The resulting factor, $\zeta$, is approximately 3.6 using
the 350-400 nm range, at the lowest LED dimmer setting of 1 V. This indicates that the photochemical cycling with the LED
light should be slower compared to ambient sunlight. We calculated $\zeta$ for the two LED dimmer settings (1 and 7 V), for both
summer and winter sunlight conditions, and for the two wavelength ranges. These results are shown in Table B1 in the SI. For
example, a LED-UV exposure of 10-15 min at 7 V is approximately 4-8 min sunlight exposure during summer in Los Angeles.

$$\zeta = \frac{\int_{\lambda_1}^{\lambda_2} E_{QF}(\lambda)_{sun} \cdot d\lambda}{\int_{\lambda_1}^{\lambda_2} E_{QF}(\lambda)_{LED} \cdot d\lambda} \qquad (2)$$

### 2.5 Kinetic model of reoxidation in the $Fe^{III}(Cit)/CA$ system

A kinetic box model (Kintecus solver package V2021, Ianni (2017)) was used to simulate the photochemical reduction and
reoxidation of iron under variable RH, both with and without copper. Kintecus is a homogeneous phase-box model, which
means that it does not account for inhomogeneity within particles. The model includes reactions, species description, and
parameter descriptions designed to replicate the flow conditions in the STXM environmental cell. A summary of the chemical
mechanisms, equilibrium constants and reaction rate coefficients used in the model is presented in Tables 1, 2 and 3. Except for
reaction R19, where the rate coefficient was adjusted from $k_r = 3.00 \pm 0.70$ to $1.50 \, M^{-1} \, s^{-1}$ for better simulation performance,
all reaction rate constants were taken from Gonzalez et al. (2017). The initial conditions for the chemical concentrations were
calculated as detailed in Section E1 of the SI, with the initial $\beta$ before UV irradiation based on STXM/NEXAFS experimental
data. A constant acidity of pH = 2 was maintained.

The phase transfer of a molecule (M) from the gas to the aqueous phase, $M(g) \longrightarrow M(aq)$, was modeled with the rate
$k_{M(g) \rightarrow (aq)} = 1/4 \omega_M \gamma_M A_p/V_p$, where $\omega_M$ is the mean thermal velocity of M, and $A_p/V_p$ is the surface to volume ratio of a
particle. The reverse rate, $M(aq) \longrightarrow M(g)$, was modeled using Henry's law and given as
$k_{M(aq) \rightarrow (g)} = k_{M(g) \rightarrow M(aq)}/(RTH_M)$. Chemical reactions and molecular diffusion can limit changes in particle composition
over time (Berkemeier et al., 2013), often leading to lower concentrations of species in the gas phase within particles than
expected in equilibrium. In the case of rapid $O_2$ uptake and reaction with condensed phase molecules (reaction R2 in Table
3), $O_2$ maintains Henry's law equilibrium only near the particle surface, while its concentration rapidly decreases due to its
reaction with abundant organic radicals. As our box model is non-depth resolving, the transfer of oxygen to the particle, $\Gamma_{O_2}$,
was parameterized to account for this limitation. We treated $\Gamma_{O_2}$ as an effective uptake coefficient parameterized as $\Gamma_{O_2} =$
$a_0 + a_1(1 - e^{-RH/a_2})$, where $a_0 = -20.0$, $a_1 = 12.40$ and $a_2 = 27.78$ based on fitting our results to those of the multilayer





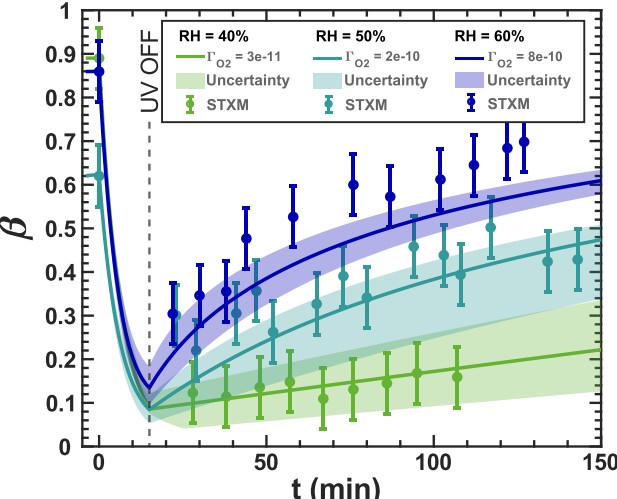

**Figure 3.** Iron(III) fraction, $\beta$, as a function of time, $t$, for 15 min of irradiation ($j = (1.4 \pm 0.2) \times 10^{-3}\,\mathrm{s}^{-1}$) followed by a period of dark reoxidation under three humidity conditions. Each humidity scenario with a different oxygen uptake coefficients, $\Gamma_{O_2}$), is shows as green for RH = 40% and $\Gamma_{O_2} = (3 \pm 2) \times 10^{-11}$, turquoise for RH = 50% and $\Gamma_{O_2} = (2 \pm 1) \times 10^{-10}$, and blue for RH = 60% with $\Gamma_{O_2} = (8 \pm 6) \times 10^{-10}$). The circles with error bars are previous data (Dou et al., 2021), while the straight lines correspond to the output from the updated Kintecus model. The model uncertainty is the shaded area stemming from uncertainty in measured RH and light intensity.

PRAD model (Dou et al., 2021). Although our box model does not account for molecular diffusion within particles, despite prior evidence of radial concentration gradients in $Fe^{III}(Cit)$, it still serves as a useful tool to explore mechanistic scenarios, particularly how copper affects $Fe^{III}(Cit)$ photochemistry. In future studies, a multilayer model informed by this work could be developed for a more detailed investigation.

Figure 3 shows previously published data from three separate reoxidation experiments with $Fe^{III}(Cit)$ : CA particles at a 1 : 1 mole ratio, conducted at RH = 60%, 50% and 40% as blue, turquoise, and green, respectively (Dou et al., 2021). Data points at $t = 0\,\mathrm{min}$ are measurements taken before UV irradiation and serve as the initial $\beta$ values for each box model scenario. The box model results shown as lines agree well with the experimental data, taking into account the uncertainties in both the experimental data and the model, which mainly stems from variations in RH at $\pm 2\%$ and light intensity. Using these

results as a reference for the $Fe^{III}(Cit)$ system, we will adapt our box model in the results section to simulate the reduction and reoxidation of photochemically aged $Fe^{III}(Cit)/Cu^{II}(HCit)/CA$ particles and explore potential reaction mechanisms. The model uncertainties (colored area) stem from the uncertainty in RH of $\pm 2.5\%$, the light intensity, and the the oxygen uptake coefficient ($\Gamma_{O_2}$).





**Table 1.** Chemical equilibria with equilibria constants ($K_{eq}$) and reaction rate constants ($k_r$) for forward and reverse reactions at pH = 2.

| Nr. | Reactions | $k_r$ | $K_{eq}$ | References |
|---|---|---|---|---|
| E1 | $H_2O \longrightarrow HO^- + H^+$ | $2.34 \times 10^{-5}\,s^{-1}$ | $1.00 \times 10^{-14}\,M$ | Welch et al. (1969) |
| E1b | $HO^- + H^+ \longrightarrow H_2O$ | $1.30 \times 10^{11}\,M^{-1}\,s^{-1}$ | | |
| E2 | $H_2O_2 \longrightarrow HO_2{}^\bullet + H^+$ | $1.26 \times 10^{-2}\,s^{-1}$ | $1.26 \times 10^{-12}\,M$ | De Laat and Le (2005) |
| E2b | $HO_2{}^- + H^+ \longrightarrow H_2O_2$ | $1.00 \times 10^{10}\,M^{-1}\,s^{-1}$ | | |
| E3 | $HO_2{}^\bullet \longrightarrow O_2{}^{\bullet -} + H^+$ | $8.00 \times 10^5\,s^{-1}$ | $1.6 \times 10^{-5}\,M^{-1}$ | Bielski et al. (1985) |
| E3b | $O_2{}^{\bullet -} + H^+ \longrightarrow HO_2{}^\bullet$ | $5.00 \times 10^{10}\,M^{-1}\,s^{-1}$ | | |
| E4 | $C_6H_8O_7 \longrightarrow C_6H_7O_7{}^- + H^+$ | $7.50 \times 10^6\,s^{-1}$ | $7.50 \times 10^{-4}\,M$ | Martell and Smith (1982) |
| E4b | $C_6H_7O_7{}^- + H^+ \longrightarrow C_6H_8O_7$ | $1.00 \times 10^{10}\,M^{-1}\,s^{-1}$ | | |
| E5 | $C_6H_7O_7{}^- \longrightarrow C_6H_6O_7{}^{2-} + H^+$ | $1.70 \times 10^5\,s^{-1}$ | $1.70 \times 10^{-5}\,M$ | Martell and Smith (1982) |
| E5b | $C_6H_6O_7{}^{2-} + H^+ \longrightarrow C_6H_7O_7{}^-$ | $1.00 \times 10^{10}\,M^{-1}\,s^{-1}$ | | |
| E6 | $C_6H_6O_7{}^{2-} \longrightarrow C_6H_5O_7{}^{3-} + H^+$ | $4.00 \times 10^3\,s^{-1}$ | $4.00 \times 10^{-7}\,M$ | Martell and Smith (1982) |
| E6b | $C_6H_5O_7{}^{3-} + H^+ \longrightarrow C_6H_6O_7{}^{2-}$ | $1.00 \times 10^{10}\,M^{-1}\,s^{-1}$ | | |
| E7 | $Fe^{3+} + C_6H_5O_7{}^{3-} \longrightarrow FeC_6H_5O_7$ | $1.00 \times 10^{10}\,M^{-1}\,s^{-1}$ | $1.58 \times 10^{13}\,M$ | Dou et al. (2021) |
| E7b | $FeC_6H_5O_7 \longrightarrow Fe^{3+} + C_6H_5O_7{}^{3-}$ | $6.33 \times 10^{-4}\,s^{-1}$ | | |
| E8 | $Fe^{3+} + C_6H_6O_7{}^{2-} \longrightarrow FeC_6H_6O_7{}^+$ | $1.00 \times 10^{10}\,M^{-1}\,s^{-1}$ | $2.51 \times 10^7\,M^{-1}$ | Dou et al. (2021) |
| E8b | $FeC_6H_6O_7{}^+ \longrightarrow Fe^{3+} + C_6H_6O_7{}^{2-}$ | $3.98 \times 10^2\,s^{-1}$ | | |
| E9 | $Fe^{2+} + C_6H_6O_7{}^{2-} \longrightarrow FeC_6H_6O_7$ | $1.00 \times 10^{10}\,M^{-1}\,s^{-1}$ | $1.94 \times 10^{10}\,M^{-1}$ | Dou et al. (2021) |
| E9b | $FeC_6H_6O_7 \longrightarrow Fe^{2+} + C_6H_6O_7{}^{2-}$ | $5.17 \times 10^{-1}\,s^{-1}$ | | |
| E10 | $Fe^{2+} + C_5H_4O_5{}^{2-} \longrightarrow FeC_5H_4O_5$ | $1.00 \times 10^{10}\,M^{-1}\,s^{-1}$ | $2.00 \times 10^3\,M^{-1}$ | Dou et al. (2021) |
| E10b | $FeC_5H_4O_5 \longrightarrow Fe^{2+} + C_5H_4O_5{}^{2-}$ | $5.00 \times 10^6\,s^{-1}$ | | |



**Table 2.** Table 1 continued.

| Nr. | Reactions | $k_r$ | $K_{eq}$ | References |
|---|---|---|---|---|
| E11 | $^\bullet C_5H_5O_5{}^{2-} + 2\,H^+ \longrightarrow {}^\bullet C_5H_7O_5$ | $1.00 \times 10^{10}\,\mathrm{M^{-1}\,s^{-1}}$ | $2.00 \times 10^3\,\mathrm{M^{-1}}$ | Dou et al. (2021) |
| E11b | $^\bullet C_5H_7O_5 \longrightarrow {}^\bullet C_5H_5O_5{}^{2-} + 2\,H^+$ | $6.67 \times 10^3\,\mathrm{s^{-1}}$ | | |
| E12 | $C_5H_4O_5{}^{2-} + 2\,H^+ \longrightarrow C_5H_6O_5$ | $1.00 \times 10^{10}\,\mathrm{M^{-1}\,s^{-1}}$ | $2.00 \times 10^3\,\mathrm{M^{-1}}$ | Dou et al. (2021) |
| E12b | $C_5H_6O_5 \longrightarrow C_5H_4O_5{}^{2-} + 2\,H^+$ | $6.67 \times 10^3\,\mathrm{s^{-1}}$ | | |
| E13 | $Fe^{3+} + H_2O \longrightarrow FeOH^{2+} + H+$ | $4.7 \times 10^4\,\mathrm{M^{-1}\,s^{-1}}$ | $1.10 \times 10^{-4}\,\mathrm{M^{-1}}$ | Brandt and van Eldik (1995) |
| E13b | $FeOH^{2+} + H^+ \longrightarrow Fe^{3+} + H_2O$ | $4.3 \times 10^8\,\mathrm{s^{-1}}$ | | |
| E14 | $FeOH^{2+} + H_2O \longrightarrow [FeOH_2]^+ + H^+$ | $1.10 \times 10^3\,\mathrm{M^{-1}\,s^{-1}}$ | $1.40 \times 10^{-7}\,\mathrm{M^{-1}}$ | Hemmes et al. (1971) |
| E14b | $[FeOH_2]^+ + H^+ \longrightarrow FeOH^{2+} + H_2O$ | $8.00 \times 10^9\,\mathrm{s^{-1}}$ | | |

# 3 Results and Discussion

## 3.1 Iron(III) reduction experiments

Figure 4 shows $\beta$ under vacuum conditions (without $O_2$ and RH = 0). Light exposure resulted in the reduction of iron from its initial condition, both in the presence and absence of copper as $CuCl_2$, with a photolysis frequency, $j_{FeCit/CuCl2}$ = $(1.4\pm0.2)\times10^{-3}\,\mathrm{s^{-1}}$. However, consistently higher values of $\beta$ were observed compared to our model predictions at longer exposure times. Our box model reproduces measurements of iron reduction in the $Fe^{III}(Cit)$/CA photochemical system, con-firming $j$ used in previous experiments conducted under vacuum conditions, with $\phi$ = 1 Dou et al. (2021). This agreement suggests that the $Fe^{III}(Cit)$/$CuCl_2$/CA system aligns with the calculated photochemical loss rate, until $\beta$ falls below 0.2 at an exposure time of 17 min. Under vacuum conditions, both $Fe^{III}(Cit)$ alone or its mixture with $CuCl_2$ are expected to be present as crystalline or amorphous (glassy) solid, potentially phase separated or mixed phase. We also applied $\phi$ = 1 in the model shown in Figure C1. From these results, we conclude the following. First, $CuCl_2$ does not affect the initial photochem-ical reaction steps to reduce $Fe^{III}(Cit)$. Second, $\phi$ for the photochemical reduction between 345-385 nm for $Fe^{III}(Cit)$/CA and $Fe^{III}(Cit)$/$CuCl_2$/CA is between 0.9-1. Previous studies have reported photoreduction of pure crystalline $Fe^{III}(Cit)$ through ob-servation of $CO_2$ production (Abrahamson et al., 1994), indicating that the solid matrix does not impede iron(III) reduction. $\phi = 0.3$ has been reported for $Fe^{III}(Cit)$ in aqueous solutions (Pozdnyakov et al., 2014). Furthermore, Corral Arroyo et al. (2022) reported enhanced photoreduction in pure $Fe^{III}(Cit)$ particles through resonant nanofocusing. Although we did not im-age photoreduction hotspots (likely due to different optical properties or non-spherical shape of the mixed particles considered




**Table 3.** Chemical reactions with rate constants and photolysis frequencies used in Kintecus at pH = 2.

| Number | Reactions | $K_{eq}/k_r/\sigma$ | References |
|---|---|---|---|
| R1 | $FeC_6H_5O_7 \xrightarrow{h\nu} Fe^{2+} + {}^\bullet C_5H_5O_5{}^{2-} + CO_2$ | $2.20 \times 10^{-3}\,\mathrm{s^{-1}}$ | Dou et al. (2021) |
| R2 | ${}^\bullet C_5H_5O_5{}^{2-} + O_2 \longrightarrow C_5H_4O_5{}^{2-} + O_2{}^{\bullet-} + H^+$ | $1.00 \times 10^6\,\mathrm{M^{-1}s^{-1}}$ | Hug et al. (2001) |
| | **ROS Reactions:** | | |
| R3 | $HO_2{}^\bullet + HO_2{}^\bullet \longrightarrow H_2O_2 + O_2$ | $8.30 \times 10^5\,\mathrm{M^{-1}s^{-1}}$ | Bielski et al. (1985) |
| R4 | $HO^\bullet + HO^\bullet \longrightarrow H_2O_2$ | $5.50 \times 10^9\,\mathrm{M^{-1}s^{-1}}$ | Sehested et al. (1968) |
| R5 | $HO_2^\bullet + HO^\bullet \longrightarrow H_2O + O_2$ | $7.00 \times 10^9\,\mathrm{M^{-1}s^{-1}}$ | Sehested et al. (1968) |
| R6 | $HO^\bullet + O_2{}^{\bullet-} \longrightarrow HO^- + O_2$ | $1.10 \times 10^{10}\,\mathrm{M^{-1}s^{-1}}$ | Sehested et al. (1968) |
| R7 | $H_2O_2 + O_2{}^{\bullet-} \longrightarrow HO_2{}^\bullet + H_2O$ | $2.70 \times 10^7\,\mathrm{M^{-1}s^{-1}}$ | Christensen et al. (1982) |
| R8 | $HO_2{}^\bullet + O_2{}^{\bullet-} \xrightarrow{H^+} H_2O_2 + O_2$ | $9.70 \times 10^7\,\mathrm{M^{-1}s^{-1}}$ | Bielski et al. (1985) |
| | **Fe(II)/(III) Oxidant Reactions:** | | |
| R9 | $Fe^{2+} + O_2{}^{\bullet-} \xrightarrow{2H^+} Fe^{3+} + H_2O_2$ | $1.00 \times 10^7\,\mathrm{M^{-1}s^{-1}}$ | Rush and Bielski (1985) |
| R10 | $Fe^{2+} + HO_2{}^\bullet \xrightarrow{H^+} Fe^{3+} + H_2O_2$ | $1.20 \times 10^6\,\mathrm{M^{-1}s^{-1}}$ | Jayson et al. (1973) |
| R11 | $Fe^{2+} + HO^\bullet \longrightarrow FeOH^{2+}$ | $4.30 \times 10^8\,\mathrm{M^{-1}s^{-1}}$ | Christensen and Sehested (1981) |
| R12 | $Fe^{2+} + H_2O_2 \longrightarrow Fe^{3+} + HO^\bullet + HO^-$ | $76\,\mathrm{M^{-1}s^{-1}}$ | Walling (1975) |
| R13 | $Fe^{3+} + O_2{}^{\bullet-} \longrightarrow Fe^{2+} + O_2$ | $1.50 \times 10^8\,\mathrm{M^{-1}s^{-1}}$ | Rush and Bielski (1985) |
| R14 | $FeOH^{2+} + HO_2{}^\bullet \longrightarrow Fe^{2+} + O_2 + H_2O$ | $1.30 \times 10^5\,\mathrm{M^{-1}s^{-1}}$ | Ziajka et al. (1994) |
| R15 | $FeOH^{2+} + O_2{}^{\bullet-} \longrightarrow Fe^{2+} + O_2 + HO^-$ | $1.50 \times 10^8\,\mathrm{M^{-1}s^{-1}}$ | Rush and Bielski (1985) |
| R16 | $[FeOH_2]^+ + O_2{}^{\bullet-} \longrightarrow Fe^{2+} + O_2 + 2\,HO^-$ | $1.50 \times 10^8\,\mathrm{M^{-1}s^{-1}}$ | Rush and Bielski (1985) |
| | **Fe(II)HCit Oxidant Reactions:** | | |
| R17 | $FeC_6H_6O_7 + O_2{}^{\bullet-} \xrightarrow{H^+} FeC_6H_5O_7 + H_2O_2$ | $6.00 \times 10^5\,\mathrm{M^{-1}s^{-1}}$ | Pham and Waite (2008) (tuned) |
| R18 | $FeC_6H_6O_7 + H_2O_2 \longrightarrow FeC_6H_5O_7 + H_2O + HO^\bullet$ | $4.20 \times 10^3\,\mathrm{M^{-1}s^{-1}}$ | Gonzalez et al. (2017) |
| R19 | $FeC_6H_6O_7 + O_2 \longrightarrow FeC_6H_5O_7 + HO_2{}^\bullet$ | $1.50\,\mathrm{M^{-1}s^{-1}}$ | Gonzalez et al. (2017) (tuned) |
| R20 | $FeC_6H_6O_7 + HO_2{}^\bullet \longrightarrow FeC_6H_5O_7 + O_2$ | $4.20 \times 10^5\,\mathrm{M^{-1}s^{-1}}$ | Pham and Waite (2008) (tuned) |
| | **Organic/FeOH Photolysis:** | | |
| R21 | $FeOH^{2+} \xrightarrow{h\nu} Fe^{2+} + HO^\bullet$ | $4.51 \times 10^{-3}\,\mathrm{s^{-1}}$ | Benkelberg and Warneck (1995) |
| R22 | $[FeOH_2]^+ \xrightarrow{h\nu} Fe^{2+} + HO^\bullet + HO^-$ | $5.77 \times 10^{-3}\,\mathrm{s^{-1}}$ | Benkelberg and Warneck (1995) |
| R23 | $C_5H_6O_5 \xrightarrow{h\nu} C_4 + CO_2$ | $2.70 \times 10^{-5}\,\mathrm{s^{-1}}$ | Dou et al. (2021) |
| R24 | $C_4 \xrightarrow{h\nu} C_3 + CO_2$ | $2.70 \times 10^{-5}\,\mathrm{s^{-1}}$ | Dou et al. (2021) |
| R25 | $C_5H_6O_5 \xrightarrow{h\nu} C_3 + 2\,CO_2$ | $2.70 \times 10^{-6}\,\mathrm{s^{-1}}$ | Dou et al. (2021) |
| R26 | $C_5H_6O_5 \xrightarrow{h\nu} C_3 + C_2$ | $2.70 \times 10^{-7}\,\mathrm{s^{-1}}$ | Dou et al. (2021) |
| R27 | $C_4 \xrightarrow{h\nu} C_2 + C_2$ | $2.70 \times 10^{-6}\,\mathrm{s^{-1}}$ | Dou et al. (2021) |



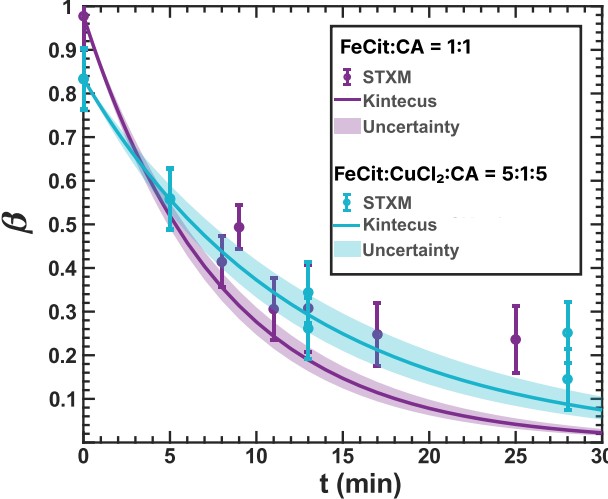

**Figure 4.** Iron(III) fraction, $\beta$, as a function of time irradiated with UV light from the LED with a narrow wavelength band centered at 370 nm. The plot shows circles and lines as measurements and model output, respectively. The light blue measurements, performed in the environmental cell without $O_2$ and RH = 0% show irradiation of $Fe^{III}(Cit):CuCl_2:CA$ at a $1:0.2:1$ mole ratio with the corresponding model output from this work. We have used UV intensity of $(2.36 \pm 0.06)\,W\,m^{-2}$, which resulted in a calculated photolysis frequency of $j = (1.4 \pm 0.2) \times 10^{-3}\,s^{-1}$ for 10 min. Violet data and model output are from a previous study using $Fe^{III}(Cit)/CA$ $(3.6 \pm 0.6)\,W\,m^{-2}$, and modeled using a photolysis frequency, $j = (2.20 \pm 0.04) \times 10^{-3}\,s^{-1}$ (Dou et al., 2021). An iron(II) quantum yield, $\phi = 1$, was used to calculate $j$. Error bars are $\pm\,0.07$ or propagated from the photon counting error, whichever is larger.

here), such photochemical enhancement may account for the higher $\phi$ in our reduction experiments compared to aqueous solutions.

## 3.2 Iron reoxidation experiments

Figure 5 shows STXM/NEXAFS images of $Fe^{III}(Cit)/CA$ particles from previous experiments (Alpert et al., 2021) (Figures 5a
225 and b), compared to images of $Fe^{III}(Cit)/Cu^{II}(Cit)/CA$ acquired in this study (Figures 5c and d), before and after reoxidation. The color bar indicates $\beta$ determined from the STXM/NEXAFS experiments. Figure 5a was acquired after 10 min of illumination followed by ~2 min of reoxidation in the dark. In this case, the smaller particles and the outermost layers of the larger particles were more oxidized compared to the centers of the larger particles. Figure 5b is a STXM image of particles on the same sample one hour after the light was turned off, showing only partial reoxidation, again, in the outermost layers. As discussed
230 previously (Alpert et al., 2021), reoxidation depends on the uptake of $O_2$, which fuels the formation of peroxy radicals that drives the process. Due to the short reacto-diffusive length for $O_2$, this reaction occurred within a depth of tens of nanometers, and the diffusion of $Fe^{II}$ from the bulk to the surface was slow (Alpert et al., 2021). When $Fe^{III}(Cit):Cu^{II}(HCit):CA$ particles were used with a mole ratio of $1:0.02:1$ (Figures 5c and d), the initial photochemical reduction was similar, with a strong




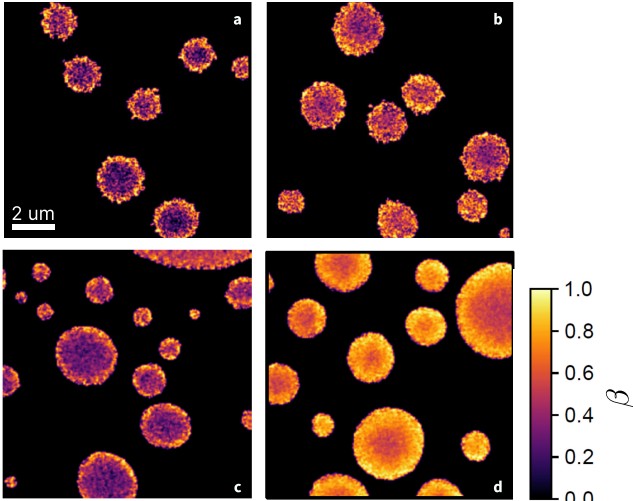

**Figure 5.** STXM/NEXAFS images of the iron(III) fraction, $\beta$, of **a** and **b** Fe$^{III}$(Cit):CA particles with a mole ratio of 1:1 shortly after irradiation and following more than three hours of reoxidation, respectively (adapted from Alpert et al. (2021) with permission by CC-BY license). **c** and **d**: Fe$^{III}$(Cit):Cu$^{II}$(Cit):CA particles with a mole ratio of 1:0.02:1 after being UV-aged for 10 min following $\sim 2$ min and one hour of reoxidation, respectively. All images were taken under the same flow cell conditions at RH = 47% and $T = 20°$C.

reduction in the particle centers and more oxidized surface layers. However, the reoxidation pattern seen in Figure 5d was different. After 1 h dark reoxidation (Figure 5d), the particles showed a substantially greater extent of reoxidation compared to Figure 5b (after 3 h in the dark), with a several hundred nm thick Fe$^{III}$ layer, a gradient in $\beta$ towards the center of the particles and significantly higher Fe$^{III}$ content throughout the particle.

The radial profile of $\beta$ as a function of pixels and distance from the perimeter and the reoxidation time, $t$, is explored in Figure 6. This figure presents $\beta$ from 1 to 35 pixels, or 35 nm to 1085 nm) from the perimeter of Fe$^{III}$(Cit):Cu$^{II}$(HCit):CA particles (mole ratio of 1:0.2:1) under RH = 47% and O$_2$, without prior irradiation. The color scale corresponds to the time since O$_2$ was introduced to the gas flow. Initial $\beta$ values of the Fe$^{III}$(Cit)/Cu$^{II}$(HCit)/CA samples were in the same range of 0.8-1 as the data of Fe$^{III}$(Cit)/CA samples published in Alpert et al. (2021) and Dou et al. (2021). As a function of O$_2$ exposure without light, $\beta$ remained constant throughout the experiment at $\sim 0.85$ within error. Only at $t > 200$ min, iron(III) was slightly reduced in the center of the particles and oxidized in the outer shells. This experiment has shown that O$_2$ alone does not cause iron(III) to significantly reduced nor oxidized. The small fraction of Fe$^{II}$ was initially likely due to handling, transferring and mounting the samples in the STXM environmental cell despite all efforts to keep them from exposure to ambient light (see Section 2.3).

In a new experiment (shown in Figure 7a), Fe$^{III}$(Cit)/Cu$^{II}$(HCit)/CA particles were irradiated for 10 min at low UV intensity, $(2.36 \pm 0.06)$ W m$^{-2}$, at the same RH = 47% as in Figure 6. A reduction in iron(III) was observed from $\beta = 0.9$ to 0.7 in the interior of the particles within the first 30 min (black dots), followed by slow reoxidization. In particular, no changes were



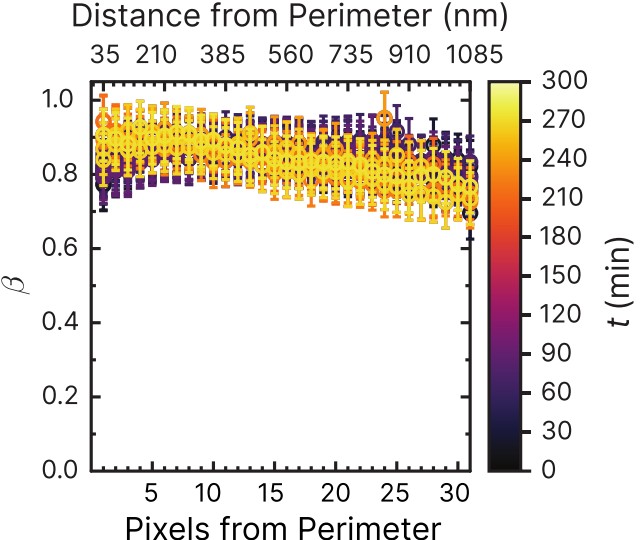

**Figure 6.** Iron(III) fraction, $\beta$, as a function of the distance and pixels from the perimeter of particles. Each line of data points corresponds to measurements of a single image of multiple particles at a given time, $t$, during $O_2$. The color scale is the $t$ after $O_2$ was introduced to environmental cell. The experiment was conducted at RH = 47% with a mole ratio of $Fe^{III}(Cit):Cu^{II}(HCit):CA = 1:0.2:1$. Error bars are $\pm$ 0.07 or propagated from the photon counting error, whichever is larger.

detected on the perimeter of the particles (pixels from perimeter $< 10$). This contrasts with a previous study without copper (Alpert et al., 2021), where $Fe^{III}(Cit)/CA$ particles exposed to similar UV intensity ($(3.60 \pm 0.06)\,\mathrm{W\,m^{-2}}$) and humidity (RH = 50%) showed markedly lower $\beta$. In that case, the initial reduction was also much more significant ($\beta \simeq 0.2$), and reoxidation occurred only slightly toward the surface. These findings indicate that copper strongly influences iron(III) reoxidation by alter-

ing the rate of $Fe^{III}$ photoreduction (R1 in Table 3), despite the comparable photoreduction observed under vacuum conditions (Figure 4).

The measurement was repeated at a higher UV intensity ($(13.55 \pm 0.06)\,\mathrm{W\,m^{-2}}$, as shown in Figure 7b). In this case, $\beta$ remained high near the surface, but decreased sharply toward the center of the particles at values $\simeq 0.4 - 0.3$. Immediate after UV lights were turned off (early reoxidation, $t$= 0-15 min), $\beta$ in the first 100 nm dropped from about 0.8 to 0.7, while $\beta$

dropped to 0.3 at 500 nm within particles. We note that $\beta$ reached values of 0.2 in the core of $Fe^{III}(Cit)/CA$ particles within the first 100 nm from the perimeter of particles (Alpert et al., 2021), in stark contrast to our observations where much higher light intensity was used. Experiments under different humidity conditions allow us to further differentiate the impact of copper on $\phi$ and the reoxidation rate.

Figure 8 shows experiments at RH = 60%, using the same mole ratios of $Fe^{III}(Cit)/Cu^{II}(HCit)/CA$ particles and irradiation

conditions as in Figure 7b. In Figure 8a, iron was seeming reoxidized immediately after being reduced. Our first measurement at $t \approx 10$ min (black dots) after light was switched off, iron was slightly reduced compared to the initial fraction in grey. This contrasts with the reduction observed in Figure 7b. We suggest two possibilities to explain this, one being that $Fe^{III}(Cit)$

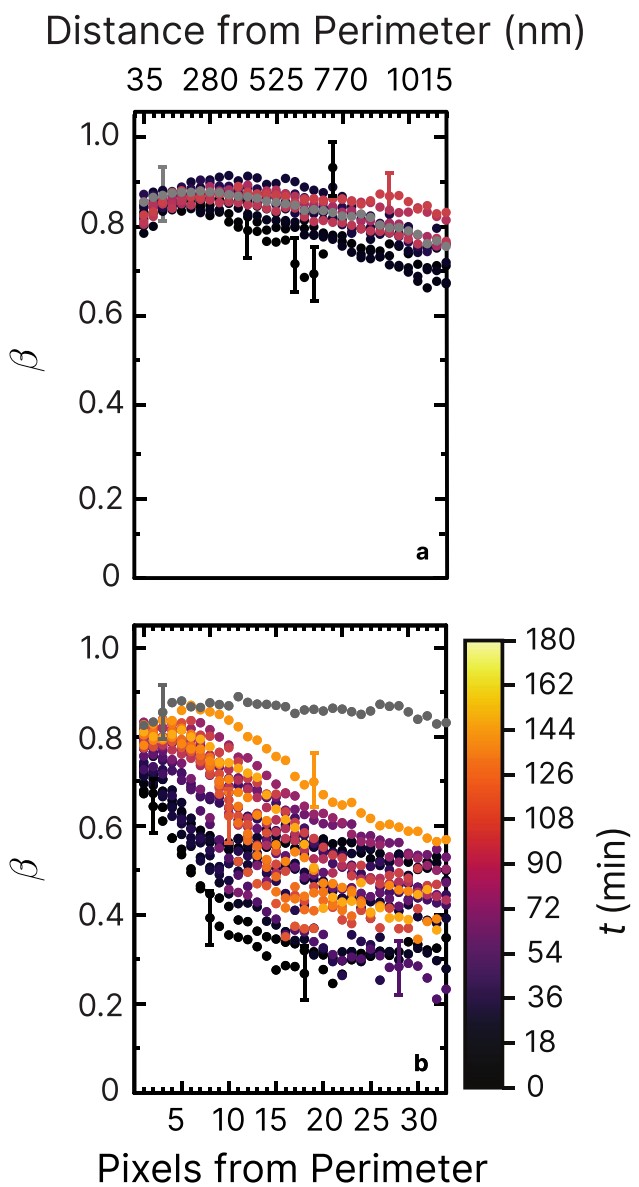

**Figure 7.** Iron(III) fraction, $\beta$, as a function of the distance and pixels from the perimeter of particles and time, $t$, after UV light was switched off at RH = 47% and a $Fe^{III}(Cit):Cu^{II}(HCit):CA$ mole ratio of $1:0.2:1$. The particles were irradiated after measuring the initial fraction $\beta$ before irradiation (gray dots). The UV intensity used was **a** $(2.36 \pm 0.06)\,W\,m^{-2}$ and **b** $(13.55 \pm 0.06)\,W\,m^{-2}$ resulting in a calculated photolysis frequency of $j = (1.4 \pm 0.2) \times 10^{-3}\,s^{-1}$ for 10 min and in **b** $j = (8.1 \pm 0.2) \times 10^{-3}\,s^{-1}$ for 10 min. The iron(III) fraction was then measured until $\approx 150\,min$ after irradiation (light orange dots). The error bars are not shown for better visibility, but similar in magnitude to Figure **??**. Error bars are $\pm 0.07$ or propagated from the photon counting error, whichever is larger.




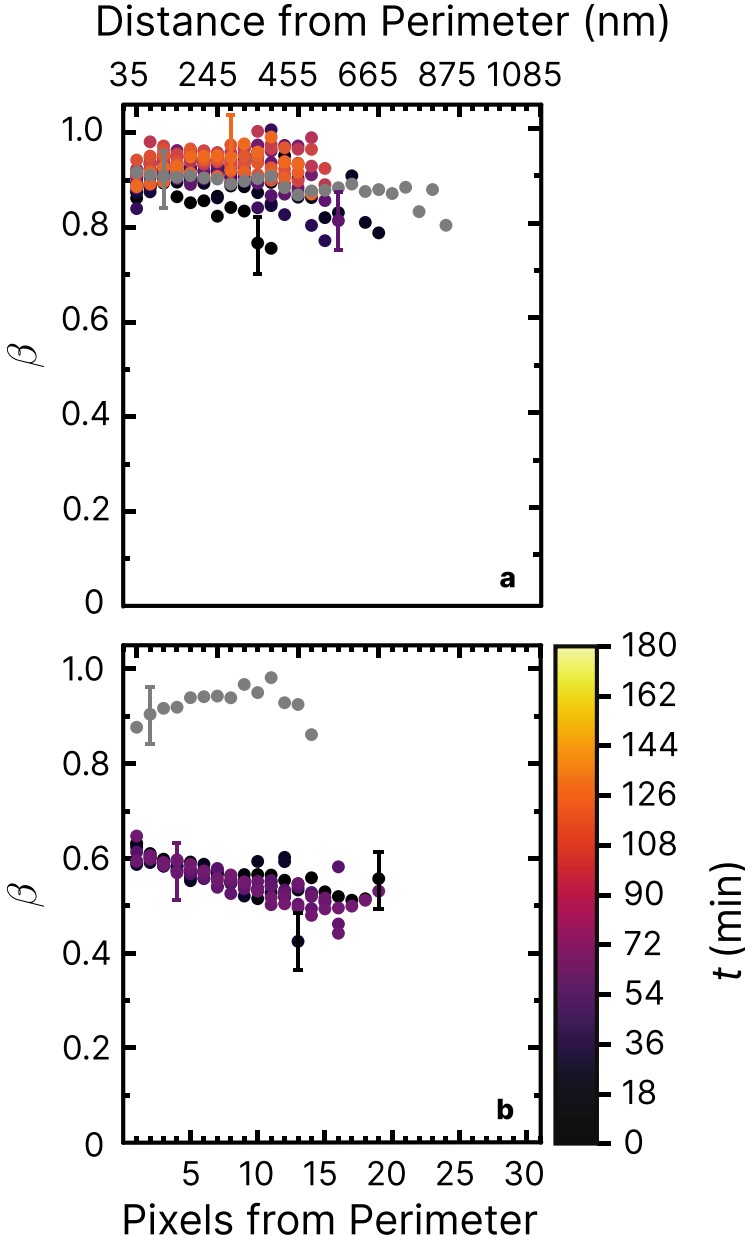

**Figure 8.** Iron(III) fraction, $\beta$, as a function of the distance and pixels from the perimeter of particles and time, $t$, after UV light was switched off at RH = 60% and a $\mathrm{Fe^{III}(Cit):Cu^{II}(HCit):CA}$ mole ratio of $1:0.2:1$ with a UV intensity of $(13.55\pm0.06)\,\mathrm{W\,m^{-2}}$ resulting in a photolysis frequency of $j = (8.1\pm0.2)\times10^{-3}\,\mathrm{s^{-1}}$. **a**: oxygen was present in the environmental cell (see Fig. 2), whereas in **b** only helium was present. Error bars are $\pm\,0.07$ or propagated from the photon counting error, whichever is larger.



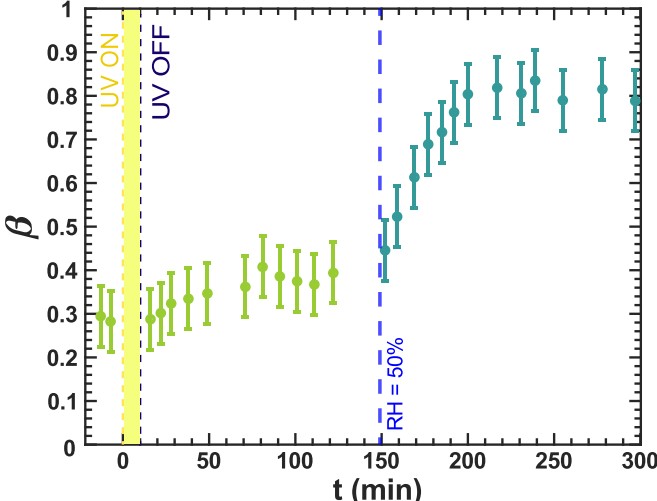

**Figure 9.** Iron(III) fraction, $\beta$, as a function of time, $t$, in a specifc experiment for irradiation at $t$ = 0-10 min at a UV intensity of $(13.55 \pm 0.06)\,\mathrm{W\,m^{-2}}$ resulting in a photolysis frequency of $j = (8.1 \pm 0.2) \times 10^{-3}\,\mathrm{s^{-1}}$ and RH = 20% with a mole ratio of $\mathrm{Fe^{III}(Cit):Cu^{II}(HCit):CA}$ = 1:0.2:1.The first data points at $t < 0$ denote the initial $\beta$ before irradiation. At $t$ = 149 min, RH was increased to 50%. Note that this experiment was conducted after the experiments in Figure 7, where irradiation occurred previously for 10 min. Error bars are $\pm$ 0.07 or propagated from the photon counting error, whichever is larger.

was not photolyzed as efficiently in the presence of copper at RH = 60%, which would mean that $\phi$ would decrease with increasing RH. Another is that copper causes a faster reoxidation, that are comparable to the time it takes to acquire the first
images after switching off light. In the experiment shown in Figure 8b, $\mathrm{Fe^{III}(Cit)/Cu^{II}(HCit)/CA}$ particles were again irradiated at RH = 60%, but without $O_2$. We observed iron was indeed reduced, reaching $\beta = 0.5$. However, this was as much as was the case for $\mathrm{Fe^{III}(Cit)/CA}$ samples at RH = 60% found to be reduced to a lower value of $\beta = 0.5$ in Alpert et al. (2021). No reoxidation was detected in Figure 8b, as we observed $\beta$ remained low for an hour. We conclude that both a low $\phi$ for iron(III) reduction (compared with $\mathrm{Fe^{III}(Cit)}$ without copper), but also faster reoxidation in presence of copper (and $O_2$) is the reason for
the observations. Experiments of $\mathrm{Fe^{III}(Cit)/Cu^{II}(HCit)/CA}$ particles containing ten times less copper (mole ratio of 1:0.02:1) is shown in the SI Figure D1, and aligns well with previous results of $\mathrm{Fe^{III}(Cit)/CA}$ (Alpert et al., 2021) and our results of $\mathrm{Fe^{III}(Cit)/Cu^{II}(HCit)/CA}$ 1:0.2:1 shown in Figure 7 and 8. The trend of $\beta$ in experiments with $O_2$ at RH = 20% (Figure D2) is similar to that at RH = 60% without $O_2$. Iron in the particles did not reoxidize at all ($\beta = 0.45$ after $t \simeq 10\,\mathrm{h}$). Thus, reoxidation here was clearly limited by the oxygen supply to the particles, probably due to diffusion limitations at high viscosity, and the
initial reduction in presence of copper was less than for $\mathrm{Fe^{III}(Cit)/CA}$.

We also assessed the impact of RH on iron reoxidation through a dynamic experiment in which RH was increased during a dark reoxidation period. In Figure 9, the iron(III) fraction averaged over all particles in an image is plotted as a function of $t$. Initially, at 20%, no reoxidation was observed, consistent with the spatially resolved images in Figure D2. At $t$ = 149 min



RH was raised to 50% and $\beta$ immediately began to increase, reaching 0.8 at approximately $198\,\mathrm{min}$. This reoxidation for
$49\,\mathrm{min}$ was much faster than in the experiment where UV irradiation occurred at RH = 47% (see Figure 7b). Reoxidation
intensified with an increase in RH, allowing $O_2$ to diffuse into the particle phase, as discussed in Alpert et al. (2021). These
results further support our conclusion that under all RH conditions, copper accelerates the rate of reoxidation of iron (II) to
iron (III) and decreases $\phi$ of $Fe^{II}$ for $Fe^{III}$(Cit) photolysis. The diffusion limitations leading to anoxic conditions (Alpert et al.,
2021) also explain our observations with $Fe^{III}$(Cit)/$Cu^{II}$(HCit)/CA particles. We aim to elucidate the role of copper in $Fe^{III}$(Cit)
photochemistry by adapting and extending the mechanism for the box model calculations in the next sections.

### 3.3   Copper containing samples revealed lower iron(II) quantum yield of $Fe^{III}$(Cit) photochemistry

The experimentally observed changes in $\phi$ in presence of copper call for its reassessment for $Fe^{III}$(Cit) with and without
copper. The experiment in Figure 4 provides evidence for $\phi = 1$ for solid $Fe^{III}$(Cit) in vacuum, and was used to explain the
reoxidation data in Figure 3 (Dou et al., 2021). Faust and Zepp (1993) investigated the photochemistry of different aqueous
iron(III) polycarboxylate complexes, including $Fe^{III}$(Cit), and determined $\phi$ as a function of $O_2$, finding a value of 0.28 in air.
Further photophysical studies using ultrafast spectroscopy revealed a long-lived iron(II) radical complex intermediate, with
various chemical fates, some of which could enable back-electron transfer (Glebov et al., 2011; Pozdnyakov et al., 2014). This
may explain a lower $\phi$ under various environmental conditions. Characterizing a series of environmentally relevant iron(III)
carboxylate complexes at multiple wavelengths in the actinic spectrum, Weller et al. (2013) confirmed a range for $\phi$ between
0.2 and 0.6. Abrahamson et al. (1994) measured $\phi$ in solution for various carboxylic acid proxies at different pH and initial
ligand-to-metal ratios. For $Fe^{III}$(Cit), they observed $\phi = 0.45$ at pH = 4 compared to $\phi = 0.28$ at pH = 2.7. This was attributed to
the formation of dimeric photoactive $Fe^{III}$(Cit) species at pH < 2, especially at high iron-to-ligand ratios, while less photoactive
monomer species dominate at pH = $0.5 - 3$ and at lower ratios. Therefore, the presence of dimeric or polymeric complexes,
formed through paramagnetic coupling among Fe centers via oxygen bridges, seems crucial for both absorbance and $\phi$. These
complexes also influence photochemical outcomes, as the reduced radical complex can interact with a remaining Fe(III) center
of the same dimer (Abrahamson et al., 1994; Glebov et al., 2011; Pozdnyakov et al., 2014). Abrahamson et al. (1994) found
photoreduction in crystalline $Fe^{III}$(Cit), but not in acetonitrile solution without water, highlighting the need for water.

Our model, simulating the reoxidation experiment at a RH = 40%, is closely aligned with the experimental results shown
in Figure 3. This indicates that $\phi$ may be influenced by the phase state of the particles, where higher viscosities immobilize
the $Fe^{II}$ radical complex, increasing $\phi$. Consequently, we modeled the low humidity experiment (RH = 40%) using $\phi$ of 0.9,
while for experiments under higher humidity (RH = 50 and 60%) and lower viscosities, we allowed $\phi$ to have a lower value.
This adjustment reflects the transition to a more dilute aqueous solution, where $\phi$ is reported mainly in the range of 0.1 to 0.4
(Abrahamson et al., 1994; Pozdnyakov et al., 2012; Weller et al., 2013). We found that a value of 0.5 led to a better fit of the
data for the two higher RH conditions than when applying $\phi = 1$, as shown in Figure 10.

Based on the experimental results presented above, copper's effect is evident in two ways. First, $\phi$ at RH $\geq$ 20% is sub-
stantially lower compared to $Fe^{III}$(Cit) alone. Second, reoxidation is accelerated in the presence of copper, resulting in near-
complete reoxidation shortly after UV light was switched off. Our box model permits changing $\phi$ with Cu content and incor-





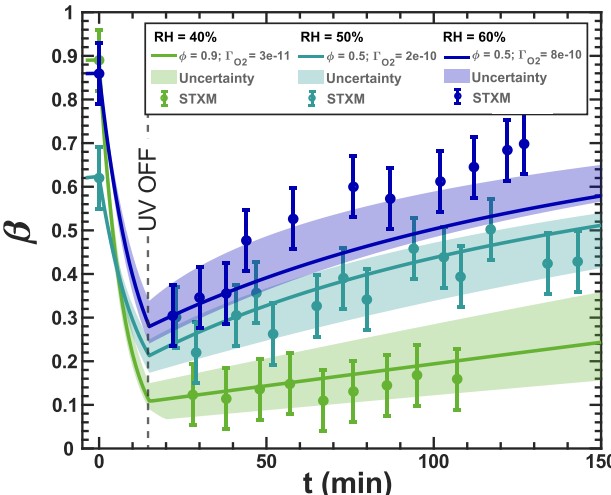

**Figure 10.** The data in Figure 3 is shown and compared with model scenarios with oxygen uptake coefficient, $\Gamma_{O_2}$, and quantum yield, $\phi$, shown as different colors (green: RH = 40% with $\phi$ = 0.9 and $\Gamma_{O_2} = (3 \pm 2) \times 10^{-11}$; turquoise: RH = 50% with $\phi$ = 0.5 and $\Gamma_{O_2} = (2 \pm 1) \times 10^{-10}$; blue: RH = 60% with $\phi$ = 0.5 and $\Gamma_{O_2} = (8 \pm 6) \times 10^{-10}$). Model uncertainties are defined as in Figure 3.

**Table 4.** Copper equilibria written as forward and back reactions with their rate constants.

| Nr. | Equilibria | $k_r$ | $K_{eq}$ | References |
|---|---|---|---|---|
| E15 | $Cu^{2+} + C_6H_5O_7{}^{3-} + H^+ \longrightarrow CuC_6H_6O_7$ | $1.00 \times 10^{10}\,M^{-1}\,s^{-1}$ | $4.79 \times 10^8\,M^{-1}$ | Perrin (1979) |
| E15b | $CuC_6H_6O_7 \longrightarrow Cu^{2+} + C_6H_5O_7{}^{3-} + H^+$ | $4.79 \times 10^{-2}\,s^{-1}$ | | |
| E16 | $Cu^{2+} + C_6H_5O_7{}^{3-} + 2H^+ \longrightarrow CuC_6H_7O_7{}^+$ | $1.00 \times 10^{10}\,M^{-1}\,s^{-1}$ | $2.19 \times 10^{11}\,M^{-1}$ | Perrin (1979) |
| E16b | $CuC_6H_7O_7{}^+ \longrightarrow Cu^{2+} + C_6H_5O_7{}^{3-} + 2H^+$ | $21.90\,s^{-1}$ | | |

porates known reactions between Cu and Fe, as well as Cu and ROS species. This extended box model will be used to fit the STXM/NEXAFS Fe$^{III}$(Cit)/Cu$^{II}$(HCit)/CA UV-aging experiments. Several studies have proposed iron-copper mechanisms for

carboxylic acid systems similar to CA (Ervens et al., 2003; Mao et al., 2013; Weller et al., 2014; Shen et al., 2021; Gonzalez et al., 2017). We integrated the relevant chemistry into the fundamental Fe$^{III}$(Cit)/CA mechanism (see Section 2.5). The resulting copper equilibria and reactions, along with their corresponding rate constants, are listed in Tables 4 and 5, and were added to the existing chemical mechanism (Tables 1, 2 and 3) in Kintecus.

Determining the iron(II) quantum yield ($\phi$) to accurately simulate the Fe$^{III}$(Cit)/Cu$^{II}$(HCit)/CA experiments required con-

ducting a reoxidation experiment in the absence of oxygen. At RH = 60% and a light intensity of $(13.55 \pm 0.06)\,\mathrm{W\,m^{-2}}$, we





**Table 5.** Copper, ROS, and Cu-Fe coupling reactions with their reaction rate constants at a constant pH = 2.

| Number | Reactions | $k_r$ | References |
|---|---|---|---|
| | **ROS Reactions:** | | |
| R28 | $Cu^+ + HO^\bullet \longrightarrow Cu^{2+} + HO^-$ | $3.00 \times 10^9 \, M^{-1} s^{-1}$ | Goldstein et al. (1992) |
| R29 | $Cu^+ + O_2 \longrightarrow Cu^{2+} + O_2^{\bullet-}$ | $4.60 \times 10^5 \, M^{-1} s^{-1}$ | Bjergbakke et al. (1976) |
| R30 | $Cu^+ + H_2O_2 \longrightarrow Cu^{2+} + HO^\bullet + HO^-$ | $7.00 \times 10^3 \, M^{-1} s^{-1}$ | Berdnikov (1973) |
| R31 | $Cu^+ + HO_2^\bullet \xrightarrow{H^+} Cu^{2+} + H_2O_2$ | $3.50 \times 10^9 \, M^{-1} s^{-1}$ | Berdnikov (1973) |
| R32 | $Cu^+ + O_2^{\bullet-} \xrightarrow{2H^+} Cu^{2+} + H_2O_2$ | $9.40 \times 10^9 \, M^{-1} s^{-1}$ | von Piechowski et al. (1993) |
| R33 | $Cu^{2+} + HO_2^\bullet \longrightarrow Cu^+ + O_2 + H^+$ | $1.00 \times 10^8 \, M^{-1} s^{-1}$ | Rabani et al. (1973) |
| R34 | $Cu^{2+} + O_2^{\bullet-} \longrightarrow Cu^+ + O_2$ | $1.00 \times 10^9 \, M^{-1} s^{-1}$ | Rabani et al. (1973) |
| | **Cu-Fe Coupling:** | | |
| R35 | $Fe^{3+} + Cu^+ \longrightarrow Cu^{2+} + Fe^{2+}$ | $1.30 \times 10^7 \, M^{-1} s^{-1}$ | Bjergbakke et al. (1976) |
| R36 | $FeOH^{2+} + Cu^+ \longrightarrow Cu^{2+} + Fe^{2+} + HO^-$ | $1.30 \times 10^7 \, M^{-1} s^{-1}$ | Sedlak and Hoigné (1993) |
| R37 | $[FeOH_2]^+ + Cu^+ \longrightarrow Cu^{2+} + Fe^{2+} + 2HO^-$ | $1.30 \times 10^7 \, M^{-1} s^{-1}$ | Mao et al. (2013) |

observed and simulated the initial reduction of iron(III) in the $Fe^{III}(Cit)/Cu^{II}(HCit)/CA$ system. Figure 11 shows the results of an optimization run, where $j$ was calculated to $(8.1 \pm 1.0) \times 10^{-4} \, s^{-1}$ using $\phi = 0.1$. In this case, oxygen-related iron(II) reoxidation was excluded, enabling a simulation solely driven by copper and/or inhibited iron(III) reduction. It is important to note that these STXM/NEXAFS data correspond to those shown in Figure 8b, but are averaged at a single time step. We

propose that $\phi_{Fe^{III}(Cit)/Cu^{II}(HCit)/CA} \ll \phi_{Fe^{III}(Cit)/CA}$ by a factor of 5 and results from copper replacing an iron center in a polynuclear complex or by inhibiting the secondary reaction of the reduced radical complex with $Fe^{III}(Cit)$ in a catalytic way.

We present the optimized simulation results for different $Fe^{III}(Cit)/Cu^{II}(HCit)/CA$ reoxidation experiments conducted under various mole ratios, RH conditions and using $\phi = 0.1$ shown in Figures 12 and 13. At RH = 20% in Figure 12, the reduction occurred while the UV lights were on. After turning them off, we observed no reoxidation, as expected, and the particles were

likely highly viscous and therefore lacked the sufficient oxygen supply needed for reoxidation, similar to the observations in Figure 11. The model is in excellent agreement with the observations considering the experimental uncertainty. At RH = 47%, reoxidation after UV lights were switched off was consistent and significant, increasing from approximately $\beta = 0.4$ to 0.8 from $t = 10$-$200 \, min$. The increased molecular diffusion at this higher RH likely accounts for the greater reoxidation compared to RH=20%. However, there is some data scatter, especially for $t = 10$-$40 \, min$, which we suggest indicates fast

reoxidation occurring on a time scale comparable to the acquisition of STXM/NEXAFS images, even under UV irradiation ($t < 10 \, min$). In particular, our model captures all the experimental data, taking into account both the model and experimental uncertainties resulting from the measurement error of RH and the light intensity, as well as the error in $\Gamma_{O_2}$. The width of the uncertainty of the model is much larger than that of other RH in Figure 12, indicating that the photochemical reduction and dark reoxidation rates are highly sensitive at RH=47%. At RH = 60%, reduction was not observed with $\beta > 0.85$ observed





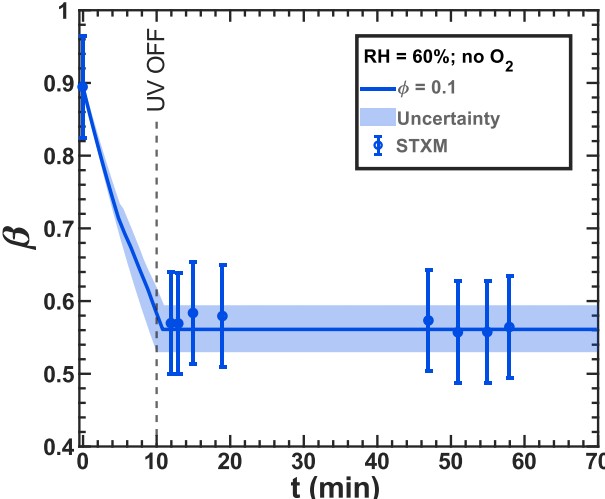

**Figure 11.** Iron(III) fraction, $\beta$, as a function of time, $t$, for irradiation at $t = 0\text{-}10\,\mathrm{min}$ with a light intensity of $(13.55 \pm 0.06)\,\mathrm{W\,m^{-2}}$ and RH = 60% with a mole ratio of $\mathrm{Fe^{III}(Cit)\!:\!Cu^{II}(HCit)\!:\!CA} = 1\!:\!0.02\!:\!1$ and without oxygen in the gas phase. The circles with error bars represent the average of all particles mapped in one time step during the STXM experiments (data of Figure 8b). Data uncertainty (error bars), model prediction and model uncertainties (straight lines and shading, respectively) are as defined in Figure 3. The iron(II) quantum yield, $\phi = 0.1$, was fit to the STXM/NEXAFS data and used to derive the photolysis frequency, $j = (8.1 \pm 1.0) \times 10^{-4}\,\mathrm{s^{-1}}$ used in the model.

throughout the experiment, possibly due to reoxidation that occurs on short time scales that we could not experimentally resolve ($< 2\,\mathrm{min}$). Despite using $\Gamma_{\mathrm{O_2}} = 1 \times 10^{-8}$, which corresponds to well-mixed conditions in the aqueous phase without any diffusion limitations, our model could not reproduce this fast reoxidation.

Figure 13 shows experimental and modeling results for a copper content reduced by a factor of ten compared to Figure 12. At RH = 47%, we again observed a reduction during irradiation by UV light and reoxidation in the dark after UV light was

switched off. Our model is in agreement with the observations, which were toward the top of the green-shaded area, indicating the model uncertainty. Unlike Figure 12, there was very little data scatter in the 30 min after UV light was switched off. At RH = 60%, we observed some reduction and reoxidation, in contrast to Figure 12. Although the data tend to scatter more for this experiment, there was a clear drop of $\beta$, from 0.8 initially to 0.6, followed by reoxidation to a maximum of 0.86 at $t$ =92 min. Our model could not capture the reduction by UV light and predicted a slower reoxidation for lower copper content.

Interestingly, in Figure 13, $\beta > 0.8$ was not reached by our model and occurred only at the last data point, while this was easily observed in Figure 12 at the same RH. From all of these observations, we suspect that a missing iron(II) reoxidation pathway exists during UV irradiation and/or during dark periods that are related to copper increasing a chemical reoxidation pathway.





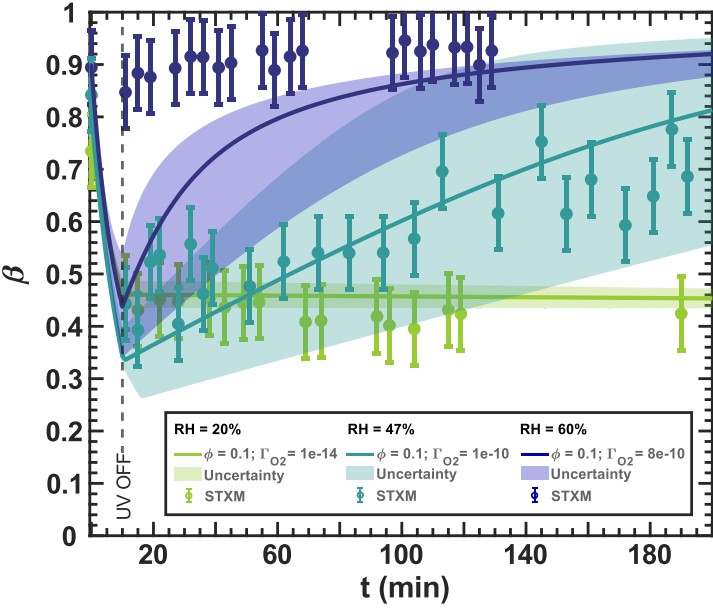

**Figure 12.** Iron(III) fraction, $\beta$, as a function of time, $t$, for irradiation at $t = 0\text{-}10\,\mathrm{min}$ with a light intensity of $(13.55 \pm 0.06)\,\mathrm{W\,m^{-2}}$, RH = 60% (blue), 47% (turquoise) and 20% (green) with a mole ratio of $\mathrm{Fe^{III}(Cit):Cu^{II}(HCit):CA = 1:0.2:1}$. The circles with error bars represent the average of all particles mapped in one time step during the STXM experiments. Data uncertainty (error bars), model prediction and model uncertainties (straight lines and shading, respectively) are as defined in Figure 3. The iron(II) quantum yield, $\phi$=0.1, was taken from the anoxic experiment shown in Figure 11, which resulted in a photolysis frequency of $j = (8.1 \pm 1.0) \times 10^{-4}\,\mathrm{s^{-1}}$). Oxygen uptake coefficients, $\Gamma_{O_2}$, are indicated in the legend.

### 3.3.1 Copper(II) oxidizing Iron(II)?

Observed reoxidation during UV irradiation and after UV light was turned off ($t = 0-40\,\mathrm{min}$) at RH = 47 and 60% (Figures 12
and 13) was underestimated. To elucidate the reasons, we derived turnover rates (d[reaction]/dt) to identify the main chemical reactions responsible for both $\mathrm{Fe^{III}(Cit):CA}$ and $\mathrm{Fe^{III}(Cit):Cu^{II}(Cit):CA}$ experiments. We selected experiments carried out at RH = 60%, where transport limitations may play a lesser role. Figures 14 and 15 show turnover rates of the $\mathrm{Fe^{III}(Cit):}$ $\mathrm{Cu^{II}(HCit):CA = 1:0.2:1}$ (RH = 60%) experiment from Table 3 ($\mathrm{Fe^{2+}/Fe^{II}(HCit)}$ oxidant reactions) and Table 5 (Cu-ROS and Cu-Fe coupling reactions), respectively. In the basic $\mathrm{Fe^{III}(Cit)}$ photochemical system (Table 3) the two dominating reactions
that oxidize iron(II) to iron(III) are R19 and R20 (green and light blue in Figure 14) with turnover rates of $\approx 10 \times 10^{-3}\,\mathrm{M\,s^{-1}}$. In these reactions, both $O_2$ and $HO_2$ act as oxidants. In the case of $\mathrm{Fe^{III}(Cit):CA}$ (RH = 60%) experiment simulation (see Figure F1), $HO_2$ is the predominant oxidant, with R20 occurring at a similarly fast rate in presence of copper. Oxidation with $O_2$ (R19) and $H_2O_2$ (R18) are one order of magnitude slower. In absence of copper, OH remains a significant oxidant (R11).




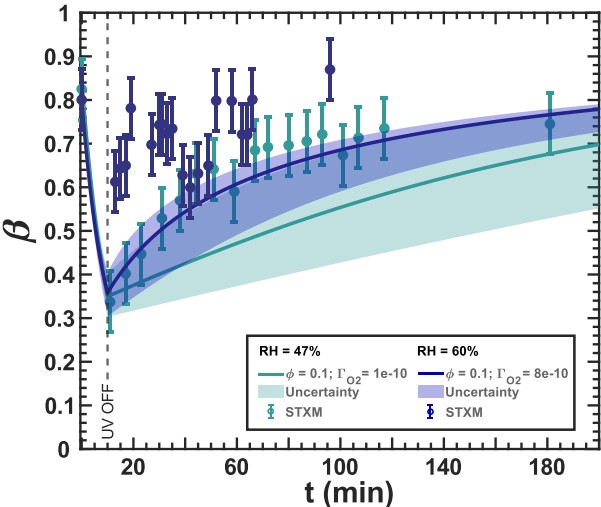

**Figure 13.** Iron(III) fraction, $\beta$, as a function of time, $t$, for irradiation at $t = 0\text{-}10$ min with a light intensity of $(13.55 \pm 0.06)\,\mathrm{W\,m^{-2}}$, RH $= 60\%$ (blue) and $47\%$ (turquoise) with a mole ratio of $\mathrm{Fe^{III}(Cit)\!:\!Cu^{II}(HCit)\!:\!CA = 1\!:\!0.02\!:\!1}$ (ten times less copper than in Figure 12). The circles with error bars represent the average of all particles mapped in one time step during the STXM experiments. Data uncertainty (error bars), model prediction and model uncertainties (straight lines and shading, respectively) are as defined in Figure 3. The iron(II) quantum yield, $\phi=0.1$, was taken from the anoxic experiment shown in Figure 11, which resulted in a photolysis frequency of $j = (8.1 \pm 1.0) \times 10^{-4}\,\mathrm{s^{-1}}$). Oxygen uptake coefficients, $\Gamma_{O_2}$, are indicated in the legend.

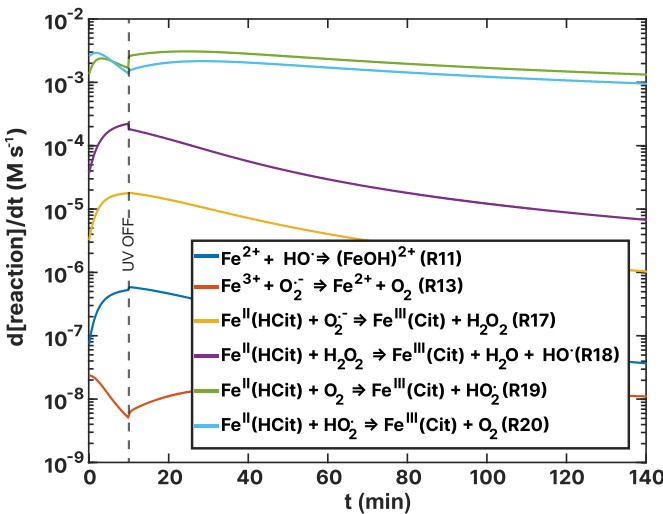

**Figure 14.** Turnover rates (d[reaction]/dt) of $\mathrm{Fe^{II}(HCit)}$ and $\mathrm{Fe^{2+}}$-$\mathrm{Fe^{3+}}$ ion loss reactions as a function of time, $t$, of the simulation of the $\mathrm{Fe^{III}(Cit)\!:\!Cu^{II}(Cit)\!:\!CA = 1\!:\!0.2\!:\!1}$ (RH = 60%) experiment as listed in Table 3. UV radiation is turned off at $t = 10$ min.



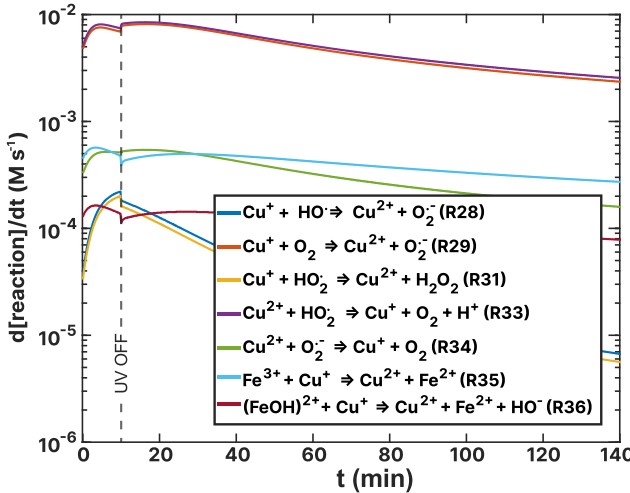

**Figure 15.** Turnover rates (d[reaction]/dt) of copper-ROS and Cu-Fe reoxidation reactions as a function of time, $t$, of the simulation of the $Fe^{III}(Cit):Cu^{II}(Cit):CA = 1:0.2:1$ (RH = 60%) experiment as listed in Table 5. UV radiation is turned off at $t = 10\,\text{min}$.

These results suggest that variations in ROS concentration and turnover may provide insight to the observed discrepancies,
which we explore below.

Among the turnover rates shown in Figure 15 for copper redox reactions, R29 and R33 are the highest, reaching $10^{-2}\,\text{M s}^{-1}$. These reveal a steady-state of hydroperoxyl radicals, where copper is initially oxidized via R29 and then $HO_2$ reduces $Cu^{2+}$ back to $Cu^+$. Notably, the only sink for hydroperoxyl radicals is the self-reaction R31 (yellow) with turnover rates between $10^{-5}$-$10^{-4}\,\text{M s}^{-1}$ producing $H_2O_2$. The concentration of $Cu^{2+}$ is sustained by $Fe^{3+}$ reoxidation as visible in light blue (R35).
We interpret these reactions as the primary contributors to elevated hydrogen peroxide ($H_2O_2$) concentration in the model of $Fe^{III}(Cit):Cu^{II}(HCit):CA$ experiments. The ROS concentrations for both $Fe^{III}(Cit):CA$ and $Fe^{III}(Cit):Cu^{II}(HCit):CA$ experiments are compared in Figure F3 in the SI. In $Fe^{III}(Cit):Cu^{II}(HCit):CA$ experiments, $H_2O_2$ concentration is the highest among all ROS at $\approx 1 \times 10^{-8}\,\text{mol L}^{-1}$. This is in contrast to $Fe^{III}(Cit):CA$ samples, where the concentrations of OH and $HO_2$ are comparable to that of $H_2O_2$, differing no more than one order of magnitude. We attribute the elevated $H_2O_2$ concentration to
the reoxidation of $Cu^+$ by $HO_2$ in R31.

Examining the $O_2$ concentrations for both samples (see Figure F4 in the SI) and comparing them with the ROS concentrations previously discussed, we observed a mildly inverse trend across all concentrations. Specifically, during UV exposure, ROS concentrations gradually rise and then consistently decline after the light is turned off. In contrast, the $O_2$ concentrations drop significantly during UV exposure and gradually increase during the dark reoxidation phase. That suggests that under UV light,
ROS production is $O_2$ limited. However, in $Fe^{III}(Cit):Cu^{II}(HCit):CA$ samples, the higher $O_2$ concentration accelerates the $Fe^{II}(HCit)$ reoxidation (R19), which is buffered by $Fe^{3+}$ reduction by $Cu^+$ in R35. The reduction of iron and the decreased hydroperoxyl radical levels in the presence of copper may explain why increasing reaction rates for R19 and R20 did not alter





**Table 6.** Tuned redox reactions incorporated into the mechanism (Tables 3 and 5.)

| Number | Reactions | $k_r$ | References |
|---|---|---|---|
| | **Cu(II) mediated Fe(II) reoxidation reactions** | | |
| R38 | $CuC_6H_6O_7 + Fe^{2+} \xrightarrow{H^+} Fe^{3+} + CuC_6H_7O_7$ | $9 \times 10^5 \, M^{-1}s-1$ | tuned |
| R39 | $Cu^{2+} + Fe^{2+} \longrightarrow Fe^{3+} + Cu^+$ | $1 \times 10^9 \, M^{-1}s-1$ | tuned |

the iron(II) reoxidation rate when attempting to fit the model to the experimental data. As a sanity check, we again increased the $O_2$ uptake coefficient to the level of well-mixed conditions, as it was done for the uncertainty limits above, to confirm that
$O_2$ flux into the condensed is not the sole driver of the fast reoxidation.

In conclusion, although not thermodynamically favored, we assert that a copper(II)-catalyzed reaction, which reoxidizes iron(II) during UV-aging, is essential for enhancing the model's performance. Matocha et al. (2005) observed a rapid reduction of $Cu^{II}$ to $Cu^I$ in the aqueous phase by dissolved $Fe^{II}$ in the absence of oxygen, under conditions where precipitation of Cu(I) oxides or hydroxides prevented reoxidation by Fe(III). Another study demonstrated synergies between iron and copper oxides
under oxygen rich conditions, suggesting the possibility of direct contact between $Fe_2O_3$ and CuO on the surface of a fly ash surrogate (Potter et al., 2018). D'Huysser et al. (1981) found similar conversion but also cautioned that they could be X-ray induced. Ammann et al. (1992) reported the conversion of mixed Cu(II)/Fe(II) into Cu(I)/Fe(III) in effloresced aerosol particles generated from mixed aqueous Cu(II)/Fe(II) chloride solutions. Additionally, Goh et al. (2006) suspected the presence of Cu(I) in chalcopyrite, indicating mixed Cu-Fe sulfides. This may suggest an interaction between copper and iron ions in
contact within mixed crystalline systems, where a different electrochemical equilibrium (favoring Cu(I)/Fe(III)) could occur, as opposed to aqueous solutions (favoring Cu(II)/Fe(II)). We hypothesize that similar coupling could take place in mixed Cu-Fe polynuclear citrate complexes. Based on this hypothesis, we included two direct reactions oxidizing Fe(II) to Fe(III) via reduction of Cu(II) to Cu(I). Although we did not explicitly model polynuclear complexes, we consider their reaction may aid in explaining our observations. The $Cu^{2+}$ induced reoxidation reactions are listed in Table 6 (R38 and R39).

The new simulation of the $Fe^{III}(Cit):Cu^{II}(HCit):CA = 1:0.02:1$ experiments with RH = 60% incorporating R38 and R39 is shown in Figure 16. The blue data points are the same as those in Figure 13. (1:0.2:1 results in Figure G1 the SI). These reactions tend to reoxidize iron at the expense of reducing Cu and agree well with our data, largely by imposing a slightly lower $O_2$ transfer coefficient of $\Gamma_{O2} = 3.5 \times 10^{-10}$, instead of $8 \times 10^{-10}$ from Figures 12 and 13). Iron reoxidation was well represented $30\,min$ after the UV light was turned off. Interestingly, the simulation exhibited transient intermediate behavior
of the iron(III) fraction with a local minimum of $\beta$ occurring at $t = 44\,min$. This is also apparent in the measurement data. A steady state $\beta$ for $t \geq 50\,min$ was also predicted in agreement with our observations. Here, the uncertainty (blue region) stemmed from $\Gamma_{O_2}$ values ranging between $1 \times 10^{-10}$ and $1 \times 10^{-9}$, with the optimal fit at $\Gamma_{O_2} = 3.5 \times 10^{-10}$. In particular, the uncertainty range in the model from $\beta = 0.4 - 1$ is much larger than our experimental error bars and points to the high sensitivity of multiphase photochemical redox reactions to $O_2$ in the condense phase as well as Cu and Fe coupled reactions
such as free iron and in an organic complex.



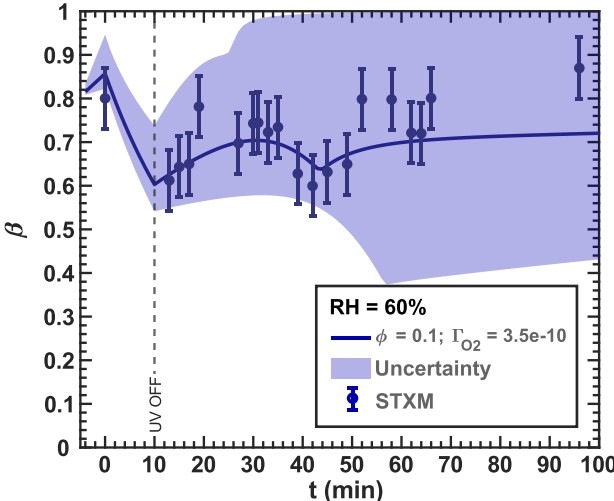

**Figure 16.** Data reproduced from Figure 13. The $O_2$ uptake coefficient of the RH =60% model run ($\Gamma_{O_2}(60\% RH)$ was slightly adjusted from $8 \times 10^{-10}$ from the initial $Fe^{III}(Cit)/CA$ mechanism to $3.5 \times 10^{-10}$. The lower and upper boundary of the model uncertainty (blue area) originate from $\Gamma_{O_2} = 1 \times 10^{-10}$-$1 \times 10^{-9}$.

To provide an explanation for these findings, we also calculated the concentrations of $O_2$, CCFR (($^\bullet C_5H_5O_5)^{2-}$) and ROS in Figures 17 and G2 in the SI. The ROS levels (Figure 17a) stabilize at a steady state concentration at $t > 40\,min$, with $H_2O_2$, decreasing to a minimum, and OH, $HO_2$ and $O_2^-$ increasing overall. The CCFR concentration did not deplete immediately after the UV light was turned off, but instead decayed from $7 \times 10^{-3}\,mol\,L-1$ at $t = 10\,min$ to zero at $\approx 40\,min$. The behavior

mirrored the concentration of $O_2$, indicating that $O_2$ was consumed by the CCFR. Reaction R39 produces $Cu^+$, which reduced $O_2$ to $O_2^-$ via R29, preventing the $O_2$ concentration from rising resulting in a slow decay of CCFR and maintaining a high value. This behavior also coincided with high $H_2O_2$ levels at $t \leq 45\,min$ due to $Cu^+$-ROS reactions R31 and R32.

The oxidation of $Fe^{2+}$ by $Cu^{2+}$ (R39) was essential to model the reoxidation of $Fe^{III}(Cit):Cu^{II}(HCit):CA$ particles under high humidity conditions. These reactions enhanced overall turnover under light, making the supply of $O_2$ an important factor.

CCFRs were a transient intermediate that strongly constrained $O_2$ reaction and diffusion to the entirety of the particle. We acknowledge that simply allowing $\Gamma_{O_2}$ to vary is not a rigorous approach to account for reaction or diffusion limitations, which may also affect other reactions. These limitations contribute to the uncertainty in $\Gamma_{O_2}$ and to the slight differences between model runs. Future work could apply a multilayer model to resolve them and explore reaction and diffusion limitations further.



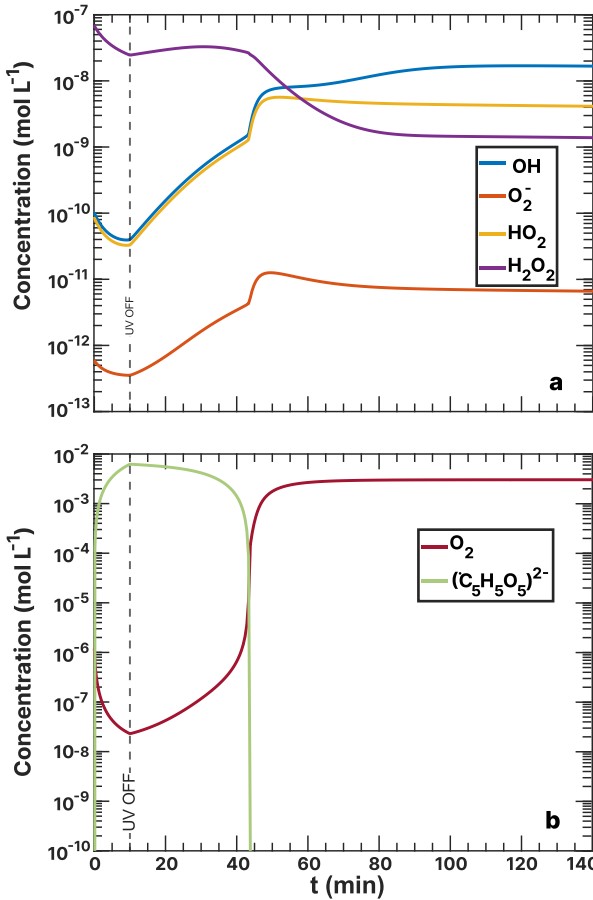

**Figure 17.** Modeled concentration of $Fe^{III}(Cit) : Cu^{II}(HCit) : CA = 1 : 0.02 : 1$ simulation including R38 and R39 from Table 6. **a** ROS concentrations. **b** Oxygen ($O_2$) and CCFR (($^\bullet C_5H_5O_5)^{-2}$) concentrations.

## 4 Conclusions

Using scanning transmission X-ray microscopy (STMX/NEXAFS), we observed novel redox characteristics in photochemically aged iron-copper complexed citric acid (CA) particles. A kinetic model was employed to explore the chemical mechanisms that led to the altered iron reoxidation pattern compared to systems without copper. The STXM/NEXAFS data provided evidence that the reoxidation of UV-light reduced iron(II) to iron(III) cause iron(III) fractions, $\beta$, to be elevated both on the surface and within the center of the particles, unlike pure iron-doped CA particles ($Fe^{III}(Cit)/CA$) (Alpert et al., 2021). Furthermore, at moderate relative humidity (RH = 60%), $\beta$ in $Fe^{III}(Cit)/Cu^{II}(HCit)/CA$ particles remained high immediately following UV irradiation.

Using a kinetic box model, we found that the changes in the photochemical aging process of $Fe^{III}(Cit)/Cu^{II}(HCit)/CA$ particles observed in anoxic STXM/NEXAFS experiments at RH = 60% originated from a lower iron(II) quantum yield ($\phi$)





of the initial $Fe^{III}$(Cit) photolysis and a faster iron reoxidation, compared to $Fe^{III}$(Cit)/CA particles. By employing the newly fitted parameter, $\phi = 0.1$, we subsequently modeled UV aging experiments in the presence of oxygen. The model simulations of low humidity experiments (RH = 20 and 47%), running on Cu-ROS and Cu-Fe coupling reactions taken from the literature, successfully replicated the experimental data. However, experiments carried out at RH = 60% using different mole ratios of copper ($M_r$ of $Fe^{III}$(Cit):$Cu^{II}$(HCit):CA = 1:0.2:1 and 1:0.02:1) could not be replicated using the same chemical mechanism.

The modeled value of $\beta$ immediately after UV irradiation was underestimated by $\approx 0.4$, reaching only $\beta = 0.9$ after $90\,\text{min}$ of reoxidation. The predicted oxygen concentration suggested that limitations in its uptake coefficient ($\Gamma_{O_2}$) could not account for the fast iron(III) reoxidation. Thus, at elevated humidity, an as yet undescribed chemical process likely accelerated the reoxidation of copper-containing particles.

Reactions between copper(II) and iron(II), reoxidizing iron(II) even during UV light exposure, proved crucial to improving
the model's performance. Previous studies have demonstrated copper-iron interactions in both solid phase (Ammann et al., 1992; Goh et al., 2006) and aqueous phase chemistry (Matocha et al., 2005). In this work, we hypothesize that such coupling may also occur in mixed Cu-Fe polynuclear citrate complexes. To test this, we included two direct reactions oxidizing iron(II) to iron(III) while reducing copper(II) to copper(I) and adjusted their rate constants accordingly. Incorporating these reactions greatly improved the simulations, and the new model runs displayed an oscillating $\beta$ pattern that began during UV aging,
closely mirroring the STXM/NEXAFS observations for more than an hour of dark reoxidation.

This study highlights the importance of understanding detailed chemical mechanisms for accurately depicting aerosol photochemical aging. Individual chemical species, such as copper, can significantly influence these processes in surprising ways, especially in organic aerosols that interact with trace compounds throughout their atmospheric lifetime. Anthropogenic emissions add further complexity to these photochemical aging mechanisms. Further work should focus on multiphase modeling
approaches that link chemistry with physical characteristics and extend these mechanisms to ambient secondary organic aerosol particles.

*Code and data availability.* The Kintecus spreadsheets for each experiment with all physical and chemical parameters used, are stored in the data inventory.





## A1    STXM/NEXAFS iron spectra

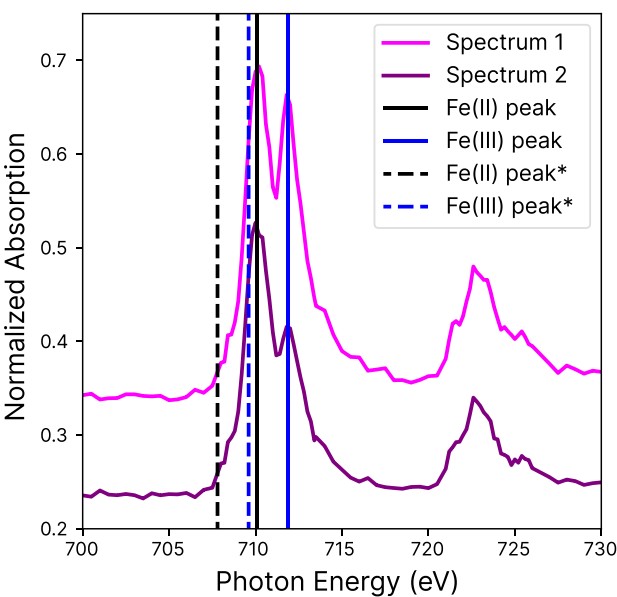

**Figure A1.** Iron arbsorption as a function of photon energies over the the L-edge with the iron(II) (black) and iron(III) (blue) absorption peaks (straight lines). The peaks were calibrated with measurements by Moffet et al. (2012) (stared) and they differed by $2.3\,\mathrm{eV}$. Two independently measured spectra are shown.

## B1    Sun actinic flux vs STXM LED

**Table B1.** Results of the actinic flux/LED flux factor ($\zeta$) with Eq. 2 for all possible combinations shown in Fig. B1.

| Sun condition | Power | Wavelength | factor, $\zeta$ |
|---|---|---|---|
| Summer LA | 1 V | 270-500 nm | 106.5 |
| Winter LA | 1 V | 270-500 nm | 57.3 |
| Summer LA | 7 V | 270-500 nm | 18.6 |
| Winter LA | 7 V | 270-500 nm | 10.0 |
| Summer LA | 1 V | 350-400 nm | 19.9 |
| Winter LA | 1 V | 350-400 nm | 10.4 |
| Summer LA | 7 V | 350-400 nm | 3.5 |
| Winter LA | 7 V | 350-400 nm | 1.8 |





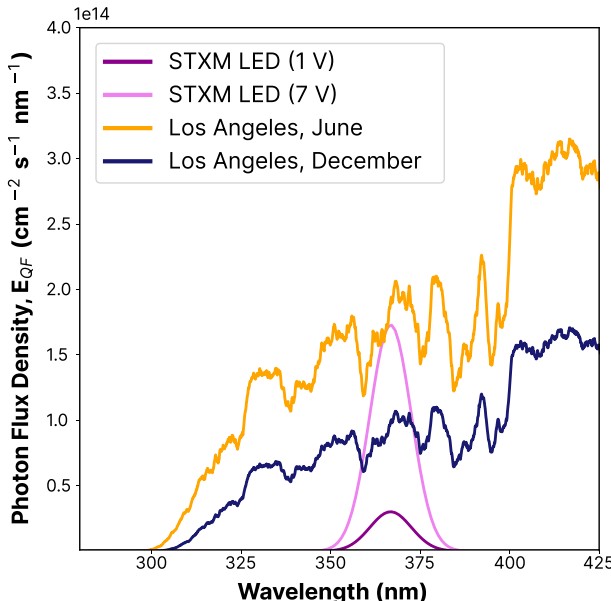

**Figure B1.** UV LED spectra flux density used for UV-aging in STXM/NEXAFS experiments for the 1 V (magenta) and 7 V setting (violet), and solar spectral flux densities in Los Angeles in June (orange) and December (dark blue) measured at noon as a function of the integrated UV-wavelength range ($\lambda = 270\text{-}500$ nm, see Eq. 2). The flux densities of the actinic spectra are taken from the *libRadtran* software Emde et al. (2016).





## C1 Iron(III) reduction with light

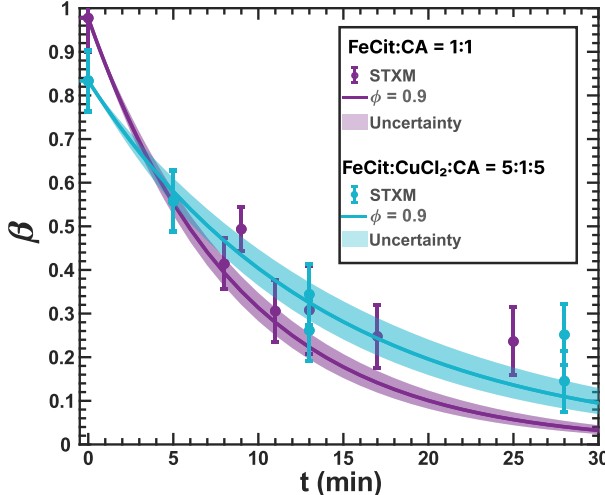

**Figure C1.** Iron(III) fraction, $\beta$, as a function of time irradiated with UV light from the LED with a narrow wavelength band centered at $370\,\mathrm{nm}$. The plot shows circles and lines as measurements and model output, respectively. Violet data and model output are from a previous study using $Fe^{III}(Cit)/CA$, and modeled using a photolysis frequency, $j = (2.2 \pm 0.2) \times 10^{-3}\,\mathrm{s}^{-1}$ (Dou et al., 2021). The light blue measurements, performed in the environmental cell without $O_2$ and RH = 0%, and model output are from this work, using $Fe^{III}(Cit):CuCl_2:$ CA at a $1:0.2:1$ mole ratio, with $j = (1.26 \pm 0.20) \times 10^{-3}\,\mathrm{s}^{-1}$. An iron(II) quantum yield, $\phi = 0.9$, was used to calculate $j$. Error bars are $\pm\,0.07$ or propagated from the photon counting error, whichever is larger.





## D1 Additional iron reoxidation experiments

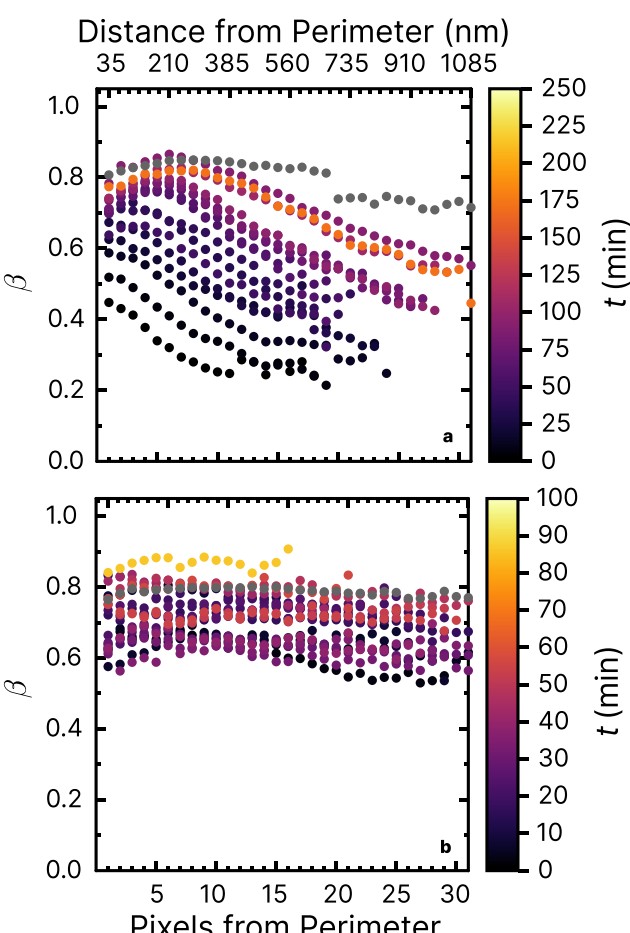

**Figure D1.** Iron(III) fraction, $\beta$, with increasing reoxidation time, $t$, after the irradiation for $10\,\mathrm{min}$ at a photolysis frequency of $j = (8.1 \pm 0.2) \times 10^{-3}\,\mathrm{s}^{-1}$ and with a mole ratio of $\mathrm{Fe^{III}(Cit)\!:\!Cu^{II}(HCit)\!:\!CA = 1\!:\!0.02\!:\!1}$. RH = 47% (a) and 60% (b). Please note the different color bars for the reoxidation time, $t$.



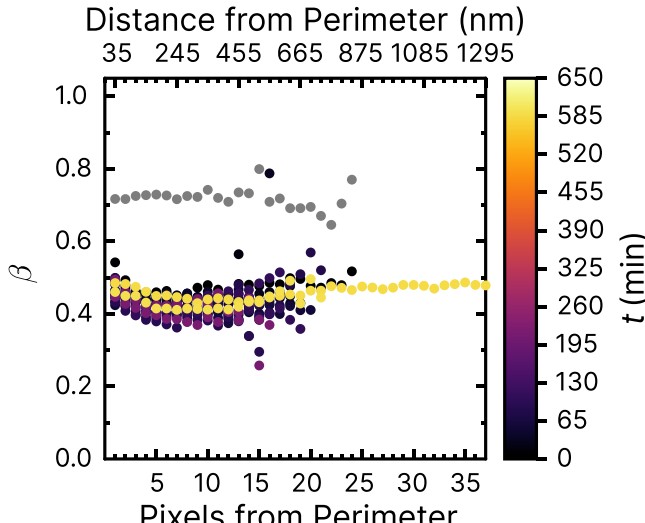

**Figure D2.** Iron(III) fraction, $\beta$, with increasing reoxidation time, $t$, after the irradiation for $10\,\mathrm{min}$ at a photolysis frequency of $j = (8.1 \pm 0.2) \times 10^{-3}\,\mathrm{s^{-1}}$ and RH = 20% with a mole ratio of $\mathrm{Fe^{III}(Cit):Cu^{II}(HCit):CA} = 1:0.2:1$.

## E1 Particle Concentration

### E1.1 density of solution

the density of the solution is dependent on the mass fraction of iron citrate and ca in the film. we start with the density of ca as a function of its mass fraction ($\rho_{ca(mfs)}$). this is measured by Lienhard et al. (2012) and fitted according to eq. E1.

$$\rho_{ca(mfs)} = 0.9971 + d_1 \cdot mfs + d_2 \cdot mfs^2 \tag{E1}$$

the $mfs$ corresponds to a certain $rh$ in terms of water activity ($a_w$) and is parameterized as follows:

$$a_w(mfs) = \frac{1 - mfs}{1 + q \cdot mfs + r \cdot mfs^2} \tag{E2}$$

an interpolation of eq. E2 gives us the exact mass fraction of ca, which we insert in eq. E1. iron citrate is not parameterized in the literature. however, **?** has determined the water activity of an aqueous $1m$ fe$^{\mathrm{iii}}$(cit) solution and found that this water activity corresponds to that of an $081m$ aqueous ca solution. to correct for this, we need to assume that everything is considered ca. we use a variable, $n_{\mathrm{cit}}$, which represents the apparent moles of citrate with the correction factor introduced above:

$$n_{\mathrm{cit}} = n_{\mathrm{ca}} + 0.81 \cdot n_{\mathrm{fecit}} + 0.81 \cdot n_{\mathrm{cucit}} \tag{E3}$$

the mass of the total solution assuming that everything is citric acid results in:

$$m_{\mathrm{cit}} = n_{\mathrm{cit}} \cdot m_{\mathrm{ca}} \tag{E4}$$





this follows in the mass of water:

$$m_{\text{h}_2\text{o}} = m_{\text{cit}} \cdot \frac{(1 - mfs_{\text{ca}})}{mfs_{\text{ca}}}, \tag{E5}$$

leading to the total mass of the solution in the flowtube (mass of the equilibrated film):

$$m_{\text{film}} = m_{\text{ca}} + m_{\text{fecit}} + m_{\text{cucit}} + m_{\text{h}_2\text{o}} \tag{E6}$$

it should be noted that these masses need to be proportionally scaled to those used to coat the flowtube. the actual mass fraction of the solute is then given by:

$$mfs_{\text{sol}} = \frac{m_{\text{ca}} + m_{\text{fecit}} + m_{\text{cucit}}}{m_{\text{ca}} + m_{\text{fecit}} + m_{\text{cucit}} + m_{\text{h}_2\text{o}}} \tag{E7}$$

the density of the equilibrated solution is calculated as a weighted mean of the dry densities times the density of ca as a function

of the mass fraction of solute ($mfs$):

$$\rho_{sol(mfs)} = \rho_{ca(mfs)} \cdot \left( \frac{n_{\text{ca}} + n_{\text{fecit}} \cdot r_{\text{fecit}} + n_{\text{cucit}} \cdot r_{\text{cucit}}}{n_{\text{ca}} + n_{\text{fecit}} + n_{\text{cucit}}} \right), \tag{E8}$$

whereas $r_{\text{fecit}}$ ($r_{\text{cucit}}$) denote the ratios between the dry iron (copper) density and the dry density of ca.

### E1.2  concentrations

the number density ($c$) of ca, fecit, cucit and h$_2$o can now be calculated with the density of the solution and its mole ratio ($a_1$,

$a_2$, ... in eq. E12):

$$c_{\text{ca}} = n_{\text{a}} \cdot \left( \frac{\rho_{\text{sol}} \cdot mfs_{\text{sol}}}{m_{\text{ca}} + a_3 \cdot m_{\text{fecit}} + a_4 \cdot m_{\text{cucit}}} \right) \tag{E9}$$

$$c_{\text{fecit}} = n_{\text{a}} \cdot \left( \frac{\rho_{\text{sol}} \cdot mfs_{\text{sol}}}{a_1 \cdot m_{\text{ca}} + m_{\text{fecit}} + a_2 \cdot m_{\text{cucit}}} \right) \tag{E10}$$

$$c_{\text{cucit}} = n_{\text{a}} \cdot \left( \frac{\rho_{\text{sol}} \cdot mfs_{\text{sol}}}{a_5 \cdot m_{\text{ca}} + a_6 \cdot m_{\text{fecit}} + m_{\text{cucit}}} \right) \tag{E11}$$

$$c_{\text{h}_2\text{o}} = n_{\text{a}} \cdot \left( \frac{\rho_{\text{sol}} \cdot (1 - mfs_{\text{sol}})}{m_{\text{h}_2\text{o}}} \right), \tag{E12}$$

where:

$$a_1 = \frac{n_{\text{ca}}}{n_{\text{fecit}}}$$
$$a_2 = \frac{n_{\text{cucit}}}{n_{\text{fecit}}}$$
$$a_3 = \frac{n_{\text{fecit}}}{n_{\text{ca}}}$$
$$a_4 = \frac{n_{\text{cucit}}}{n_{\text{ca}}}$$
$$a_5 = \frac{n_{\text{ca}}}{n_{\text{cucit}}}$$
$$a_6 = \frac{n_{\text{fecit}}}{n_{\text{cucit}}}$$




## F1    Fe$^{III}$(Cit)/CA reaction turnovers and concentration profiles

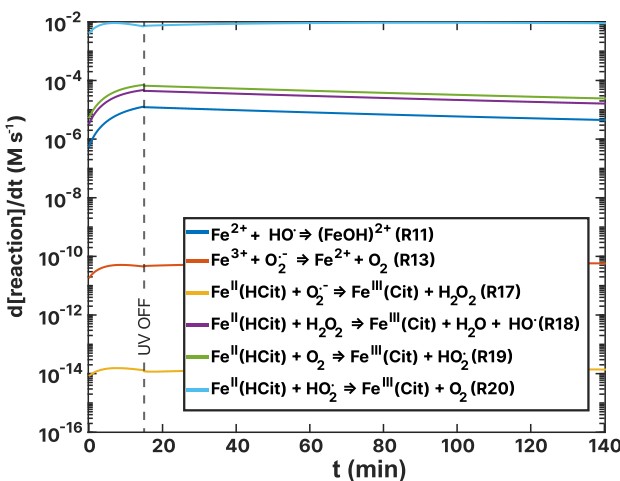

**Figure F1.** Turnover rates (d[reaction]/dt) as a function of time, $t$, of Fe$^{II}$(HCit) and Fe$^{II}$/Fe$^{III}$ ion loss reactions of the Fe$^{III}$(Cit) : CA = 1 : 10 simulation of the (RH = 60%) experiment as listed in Table 3. UV radiation is turned off at $t = 15$ min.

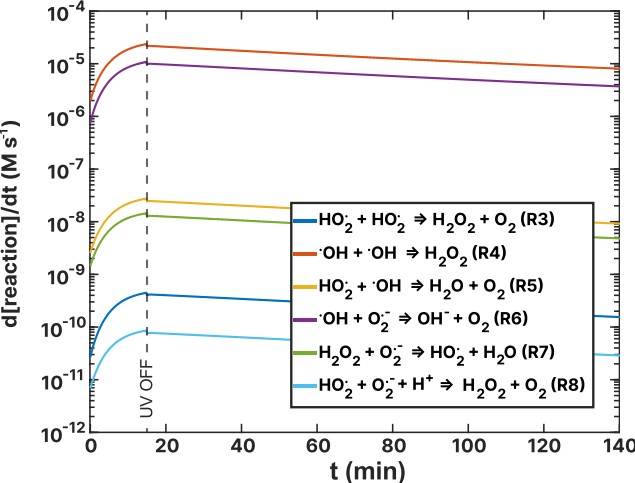

**Figure F2.** Turnover rates (d/dt) of the ROS reactions listed in Table 3 as a function of time, $t$, of the Fe$^{III}$(Cit) : CA = 1 : 10 experiment (a) and the Fe$^{III}$(Cit) : Cu$^{II}$(HCit) : CA = 1 : 0.2 : 1 experiment (b) simulations. UV radiation was turned off at $t = 15$ min.



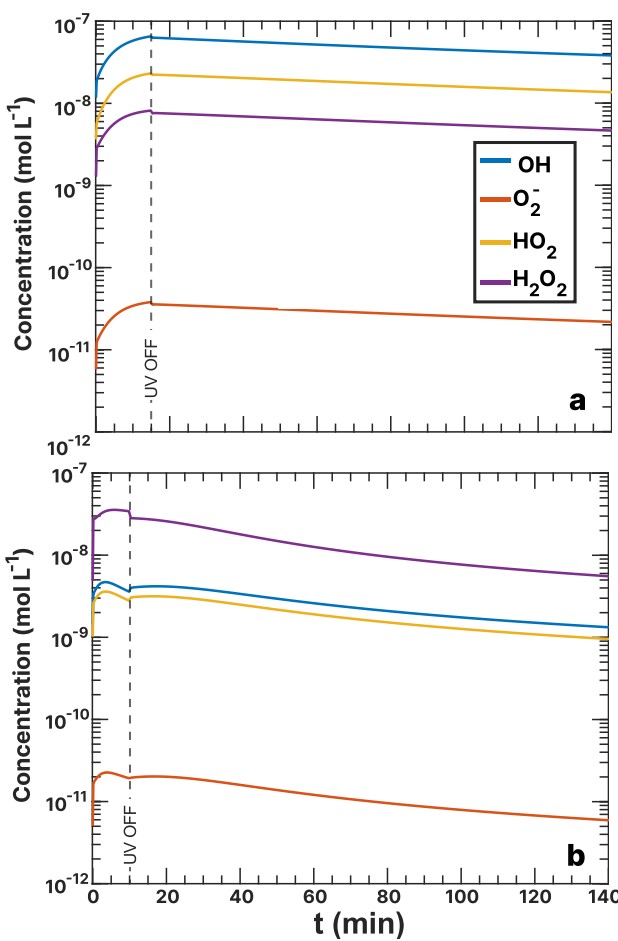

**Figure F3.** ROS concentrations during $Fe^{III}(Cit):CA = 1:10$ (**a**) and $Fe^{III}(Cit):Cu^{II}(HCit):CA = 1:0.2:1$ (**b**) experiment simulation as a function of time. UV radiation was turned off at $t = 15\,min$ (**a**) and $10\,min$ (**b**).



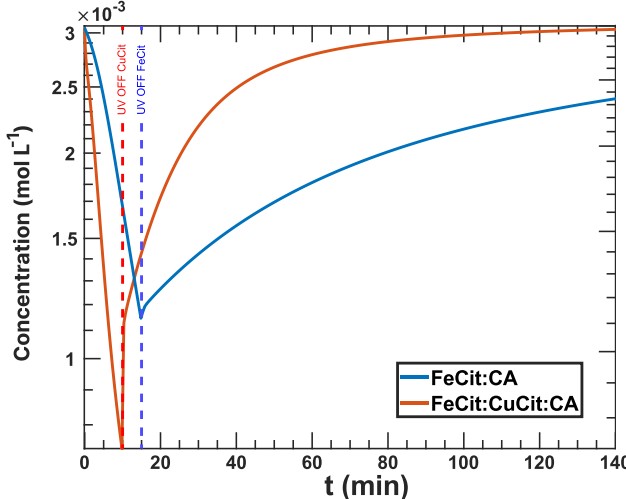

**Figure F4.** Oxygen concentrations during the $Fe^{III}(Cit):CA = 1:10$ and the $Fe^{III}(Cit):Cu^{II}(HCit):CA = 1:0.2:1$ experiment under RH = 60% simulation as a function of time. UV radiation was turned off at $t = 10\,min$ and $15\,min$, respectively.




## G1 Simulations of $Cu^{2+}$ reduced reoxidation reactions of $Fe^{III}(Cit):Cu^{II}(HCit):CA = 1:0.2:1$ samples

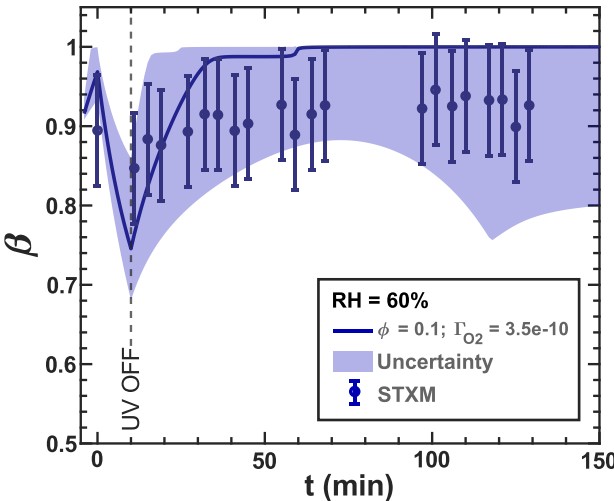

**Figure G1.** Iron(III) fraction, $\beta$, with increasing dark reoxidation time ($t$) after UV radiation ($t = 10$ min). $Fe^{III}(Cit):Cu^{II}(HCit):CA = 1:0.2:$
1 samples. The circles with error bars represent the average of all particles mapped at one time step during the STXM experiments, and the
straight lines express the Kintecus model output. The $O_2$ uptake coefficients ($\Gamma_{O_2}$) were determined from the initial $Fe^{III}(Cit)$/CA mechanism
(see Section 2.5). The lower and upper boundary of the model uncertainty (blue area) originate from $\Gamma_{O_2} = 1 \times 10^{-10}$-$1 \times 10^{-9}$. The iron(II)
quantum yield ($\phi$) was adapted from the anoxic experiment shown in Figure 11, optimized towards the new $Cu^{2+}$ induced reoxidation reaction
rates, and multiplied by the photolysis frequency ($j = (8.1 \pm 1.0) \times 10^{-3}\,s^{-1}$). The reaction rates for R38 was $2 \times 10^6\,M^{-1}s^{-1}$ and for R39
$7 \times 10^8\,M^{-1}s^{-1}$. Uncertainty in the experimental derived $\beta$ was determined as, $\Delta\beta = \pm 0.07$.



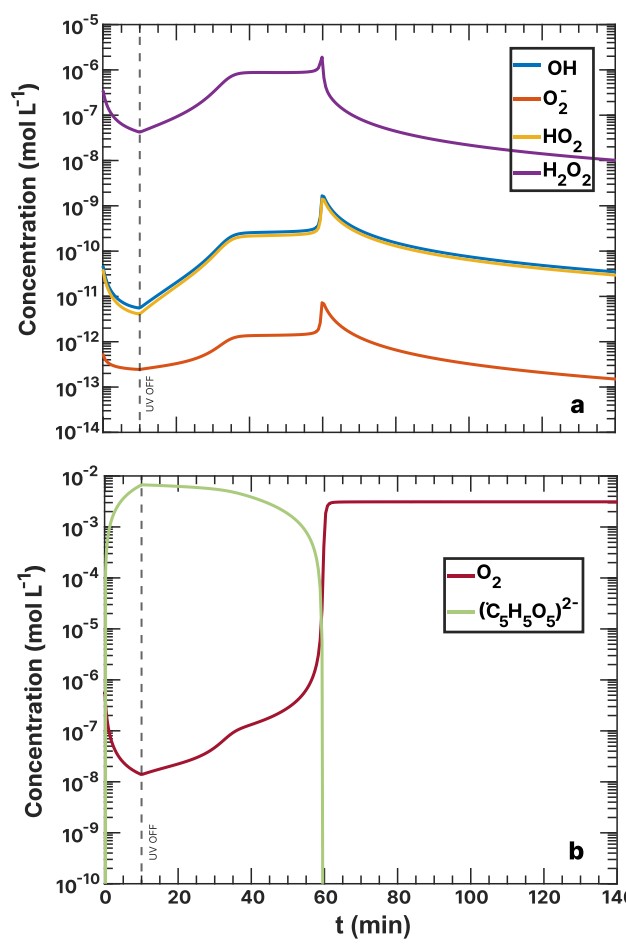

**Figure G2.** Concentration results of $Fe^{III}(Cit):Cu^{II}(HCit):CA = 1:0.2:1$ simulation including R38 and R39 from Table 6. **a** ROS concentrations. **b** Oxygen ($O_2$) and CCFR (($^\bullet C_5H_5O_5)^{-2}$) concentrations.

*Author contributions.* MA, PAP and KK designed the research. LT, RKYC, PAP and KK carried out the STXM/NEXAFS measurements. PAP and KK did the modeling work with valuable inputs by MA. KK wrote the manuscript with significant inputs by PAP and MA.

*Competing interests.* At least one of the (co-)authors is a member of the editorial board of Atmospheric Chemistry and Physics

510



*Acknowledgements.* The authors thank the Swiss National Science Foundation for financial support with grants no. 188662 and 189883. The authors thank Prof. Christian Ludwig for having facilitated this work and LT's salary was funded by the Swiss National Foundation (project no. 184817). In situ electron and X-ray absorption spectroscopy experiments were hosted at the POLLUX Beamline at the Swiss Light Source (SLS), the support of which is highly appreciated.

515       .



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
