# Peer review of "Copper accelerates photochemically induced radical chemistry of iron-containing SOA"

_EGUsphere, 2024_

## Author Comment (AC1)

**G2   Calculation of reacto-diffusive length for O$_2$**

We know from Hanson et al. (1994) that the reacto-diffusive length is defined as:

$$l_D = \sqrt{\frac{D_l}{k^I}}, \tag{G1}$$

whereas $D_l$ is the liquid phase diffusion coefficient (taken from Dou et al. (2021)) and $k^I$ describes the sum of the oxygen sinks corresponding to the turnover rate of reaction R2 in Table 3 ($3.9 \times 10^{-2}\,\mathrm{M\,s^{-1}}$) in our case divided by the O$_2$ concentration in steady state ($C_{ss}$ in $moles$, taken from Figure F4). In this case, it is equivalent to:

$$k^I = \frac{k^{R2}}{C_{ss}}, \tag{G2}$$

This leads to a reacto-diffusive length, $l_D$, for oxygen of $2.66 \times 10^{-8}\,\mathrm{m}$.

*Author contributions.* MA, PAP and KK designed the research. LT, RKYC, PAP and KK carried out the STXM/NEXAFS measurements. PAP and KK did the modeling work with valuable inputs by MA. KK wrote the manuscript with significant inputs by PAP and MA.

*Competing interests.* At least one of the (co-)authors is a member of the editorial board of Atmospheric Chemistry and Physics

*Disclaimer.* TEXT

*Acknowledgements.* The authors thank the Swiss National Science Foundation for financial support with grants no. 188662 and 189883. The authors thank Prof. Christian Ludwig for having facilitated this work and LT's salary was funded by the Swiss National Foundation (project no. 184817). In situ electron and X-ray absorption spectroscopy experiments were hosted at the POLLUX Beamline at the Swiss Light Source (SLS), the support of which is highly appreciated.

.

---

## Author Response (AR1)

RC1:

Kilchhofer et al. report interesting data sets to examine the effect of the co-existence of Fe and Cu on the photochemistry of citrate-containing particles. Based on the experiments, the present study found a synergy between iron and copper in photoreactions. However, the text requires minor revisions, as addressed below:

*AC: We would like to thank the reviewer for taking the time to review our manuscript and for their positive feedback. We have revised our manuscript according to the reviewer's comments and suggestions and provide a point-by-point answer below. The reviewer's comments are given in black, our responses in italic, and the actions in the manuscript are marked with ' '. The actions are also visible in an additional 'latexdiff' file together with the submission of the revised version.*

The main text needs to mention Figure 1. It would be easier to follow if the authors had added a pathway number to Figure 1 and referred to it along with the text (lines 52-64).

*Good point. We have adapted this by putting the color of the molecules as depicted in Figure 1 to the respective sentences in the text in parenthesis. For example on line 58: '($C_5^{2-}$, orange in Figure 1)'.*

Line 102: What is "it"?

*Thank you for pointing this out. We have rephrased the sentence on line 117 to: 'In order to quantify the iron oxidation states, ...'*

Line 125: What is the size to select here?

*We have used sizes from 300-400 nm and let them agglomerate on the SiNi window during impaction. The sizes were chosen in order to get plenty of particles at a size of 0.2-1 μm on the SiNi window. Thus, the impaction time varied as a function of mobility diameter from approx. 50 to 100s. We have complemented this in the text as following on line 140: '... , following passage through a differential mobility analyzer to select a dry mobility diameter of 300-400 nm. The particles were nebulized for 50-100 s to ensure a homogeneously distributed sample with particles of 0.2-1 um in diameter.'*

What is the rationale for assuming the constant acidity of pH = 2?

*Attempts in our previous flow tube experiments (Alpert et al., 2021 .; Dou et al., 2021) to measure the pH of the coating before and after reaction indicated a pH of around 2 and no difference between before and after. We assume that for these STXM experiments the same would apply and also that Cu would not generally change this. For the model calculations, we have first allowed the proton concentration to vary. It indicated rapid fluctuations but no significant temporal evolution. The main argument for keeping it*

constant is that while the acid-base equilibrium of citric acid and of $HO_2$ and $O_2^-$ can be well represented in the model, the model does not track the products of citric acid decomposition explicitly. Therefore, we cannot keep track of carboxylic acids in the system, which likely have a strong buffering effect. We have adapted the text on line 147 as follows: 'The measurements of the acidity of the solutions did not differ between the different mole ratios and the pH was quantified to approx. 2. Because we only sampled a very low mass of particles, we were unable to measure the pH after a STXM experiment. The solution was then nebulized and the …'

Figure 5 shows two identical images (5a and b only), which need revision.

*Fig. a and b are different. Panel a represents the FeCit/CA shortly after irradiation and panel b the same sample (another window of particles) after three hours of reoxidation. Both are taken from Alpert et al., 2021. We have revised the description of the small, but still relevant differences between Fig. 5a and 5b and the context of that, along with better emphasizing the contrast seen in presence of Cu shown in Fig. 5c and 5d. The text in the manuscript is now revised from line 244 to 256.*

Lines 273-275 conclude that the observations are due to a reduced quantum yield for iron(III) and faster reoxidation in the presence of copper. As noted in the text, particle viscosity also affects the beta. It needs to be confirmed that the particle viscosity is the same among different particle compositions at a given RH.

*Thank you for pointing this out. This is indeed an interesting argument, and we hypothesize that a low concentration of an additional transition metal does not affect the viscosity significantly. We have previously directly measured the viscosity of citric acid solutions with iron(III) citrate at high water activity (Alpert et al., 2021), and for lower water activity, the diffusivity of solutes was constrained by the kinetic model (Dou et al., 2019, 2021). A more detailed analysis of the influence of copper on the viscosity of such particles at low water activity was out of scope for the present work. Indeed, in the meantime, we have started working on a follow-up project using the poke-flow technique in collaboration with Prof. Allan Bertram at University of British Columbia. This will be presented in an upcoming manuscript. We have added this explanation in section 2.2 on line 151 as follows: 'The viscosity of the particles at the same RH did not vary between different particle compositions as the concentrations of iron was too low to influence the viscosity of CA as proven by Alpert et al., 2021, while we assume that the small amount of additional copper did not change that.'*

How was the particle viscosity change at different RHs represented in the model calculation?

*This is done by using an oxygen uptake coefficient parameterized as $\Gamma O2 = a0 + a1(1 - e^{-RH/a2})$, where $a0 = -20.0$, $a1 = 12.40$ and $a2 = 27.78$ based on fitting our results to those of the multilayer PRAD model by Dou et al., 2021 as indicated in the main text on line 205 on page 8. And of course, in the present work the model otherwise did not*

*consider diffusion within the bulk phase, but only considered homogeneous bulk phase chemistry. This allowed inclusion of a much more complex chemical reaction scheme at the expense of not spatially resolving transport within the particles. We added a sentence at the end of the Methods section on line 223 to clarify this: 'This uptake coefficient was used in the model calculations to represent the change in particle viscosity a*

Does Figure 9 show the beta averaged over all particles deposited on the substrate? The authors should show the position-dependent beta values to strengthen the discussions about the O2 diffusion-limited process.

*Yes. The data in Figure 9 (t < 150 min) is the beta averaged of 10-15 particles on one SiNi window at a given reoxidation time. In other words, one data point in Figure 9 represents the average of a line in Figure 7a at one time point including 10-15 particles. We have changed the sentence from line 303 on. 'In Figure 9, the iron(III) fraction averaged over 10 to 15 particles in an image is plotted as a function of t, whereas in Figure 7 the position-dependent iron(III) fractions are shown.'*

Line 286: Please specify what is "These results". So far, I don't see which results support the acceleration and the reduced quantum yield by copper and how it works.

*Ok, and we agree. We have clarified that 'these results' refers to the different RH scenarios (Fig. 7 and 8) and not to Fig. 9, where we have changed RH during an ongoing experiment. We have changed the sentence from line 307 with the details accordingly: 'Reoxidation intensified with an increase in RH, allowing $O_2$ to diffuse into the particle phase, as discussed in Alpert et al., 2021 and in Figures 7 and 8'. Furthermore, we have shifted the discussion of the reduced quantum yield by copper to the section 3.3 on line 311 and argue that the findings of Figure 9 are consistent with Figures 7 and 8.*

The section starting with line 292 seems more like a literature review about the quantum yields, but the connection to the present study needs to be discussed sufficiently. Furthermore, how is the discussion useful for the effect of copper on the reduced quantum yields?

*We see your point. We have rephrased the first paragraph (from line 311 on) and move most of the 'literature review about the quantum yields' into the last paragraph of the Introduction (page 3 line 65). This paragraph specifically highlights the sensitivity of the iron(II) quantum yield to the presence of polynuclear iron(III) complexes. It thus better prepares for the discussion of the impact of copper on the iron(II) quantum yield in the first part of the section (around Figure 11) and the missing reoxidation pathway (based on Figures 12 and 13) in the middle part of that section. This leads to the second hypothesis of copper in the polynuclear iron complexes facilitating iron (II) reoxidation in a way similar to mixed solid Fe-Cu-oxides, sulfides or chlorides reported in previous studies, discussed in the third part of the section. Thus, we have rephrased this part of*

*the discussion section and have emphasized the role of polynuclear complexes to facilitate the way copper affects this system.*

RC2:

This is an interesting and well-executed study investigating the role of Cu in the photochemical aging of Fe-containing secondary organic aerosols. The authors present compelling experimental data showing how Cu impacts Fe(III) reduction during photochemical aging. Additionally, the study demonstrates that the results are significantly influenced by relative humidity (RH). While the findings are valuable, several aspects require further clarification and revision before I can recommend the paper for publication.

**Major Comments**

1. **Reason for Rapid Fe Reoxidation**

Figures 10, 12, and 13 indicate that the observed Fe reoxidation occurs much faster than predicted by the model at RH = 60%. The authors attribute this discrepancy to the RH dependence of Fe(II) quantum yield. However, an alternative explanation could be the underestimation of internal $H_2O_2$ or OH production in the model, especially considering Cu's role. Cu reacts with $HO_2$ (or $O_2^-$) far more efficiently than Fe in the aqueous phase, leading to increased $H_2O_2$ production (see for example Mao et al., 2017). Elevated $H_2O_2$ or OH levels could accelerate Fe(II) oxidation to Fe(III), explaining both the rapid reoxidation and the reduced Fe(III) reduction observed in the Fe(III)(Cit)/Cu(II)(HCit)/CA system compared to the Fe(III)(Cit)/CA system.

*This is correct and we are aware of possible rapid Fe reoxidation reactions with $H_2O_2$ or OH species. We have simulated all reactions from the literature we were aware of (Berdnikov, 1973; Bjergbakke et al., 1976; Goldstein et al., 1992; Mao et al., 2013, 2017; Rabani et al., 1973; Sedlak & Hoigné, 1993; von Piechowski et al., 1993) and could still not explain the rapid reoxidation.*

*Related to the question of whether we have underestimated $H_2O_2$ production, we note that in our previous work, for the iron(III) citrate system, $H_2O_2$ concentrations were constrained by the measured $HO_2$ production and by the observed mass loss and iron(II) reoxidation rate. Here, we have taken the literature value for the reaction rate coefficient for the self-reaction rate of $HO_2$ radicals (R3). Hence, we may have underestimated the total peroxide concentration, because we did not model the secondary $RO_2$ production as discussed in Alpert et al., 2021. On the other hand, our Fe only chemical mechanism did simulate the iron(II) reoxidation well with the chosen literature rate coefficients. We also note that copper essentially acts as a $HO_x$ sink, which would rather not lead to acceleration of iron(II) oxidation. Nevertheless, we have discussed these uncertainties and*

*alternative explanations for the acceleration of Fe(II) oxidation in section 3.3 from line 310 on.*

As authors state in the text, O2 is consumed by abundant organic radicals. Organic radicals formed during photochemical aging (wavelength 360–380 nm) likely produce significant amounts of $HO_2/RO_2$ radicals, contributing to additional $H_2O_2/OH$ production that the current model may not account for.

*The only $HO_2$ source in the model is elimination of $H_2O_2$ from the citrate peroxy radical, while all other pathways of secondary $RO_2$ or $HO_2$ formation are not explicitly included, as mentioned in the response to the preceding comment. Additional photolysis channels apart from iron(III) citrate photolysis, e.g., photolysis of $C_5$ ketone products, are not simulated to be $HO_2$ sources, but only contribute to mass loss. In that sense, also for this, the model had been optimized to fit to available observables. And yes, we have placed a caveat on the uncertainty about the true $HO_2$ and $H_2O_2$ production and adapted the text on line 378 as follows: 'We also emphasize that our model does not explicitly include other $HO_2$ formation pathways such as photolysis of $C_5$ ketone products other than elimination of $HO_2$ from the citrate peroxy radical, which could cause an uncertainty about the true $HO_2$ and $H_2O_2$ production. For this, the model had been optimized to fit the available observables. From all of these observations and the fact that we have included all relevant reactions involving copper in our box model, we suspect that a missing iron(II) reoxidation pathway exists during UV irradiation and/or during dark periods that are related to copper increasing a chemical reoxidation pathway.'*

Additionally, the reaction rate constant for R34 appears to be incorrect. Table 5 lists it as 1x10^(-9) /M/S, but it should be 8x10^(-9) /M/S. A lower rate constant in the model would underestimate $O_2^-$ production, leading to underestimated $H_2O_2$ levels. The authors should explore the possibility of underestimated oxidant levels in the model instead of relying solely on RH-dependent quantum yield for Fe(II), which lacks a clear scientific basis.

*Thanks for pointing this out and you were right. We have updated the model and tested the impact of a higher rate constant of R34 ($8e-9$ $M^{-1}$ $s^{-1}$). We could not see any changes in the reoxidation pattern, beta (see in the attached pd file 'Review_reaction_rate_R34.pdf'). We have also checked the sensitivity of the results to a number of other parameters by changing them upwards and downwards by about three orders of magnitude, before coming up with the hypothesis of needing a lower RH-dependent quantum yield for Fe(II) and an additional reaction between Fe(II) and Cu(II). None of these additional simulations succeeded in correctly simulating the reoxidation rate. For example, we have manually tuned R35 in order to assess the role of iron-copper (re)oxidation patterns. Only the reoxidation of iron(II) to iron(III) during the dark reoxidation phase could be changed slightly. However, none of the reactions could compete with the photolysis of iron(III)-citrate under light to bring beta back as fast as it was observed in the experiments. We note that our mechanism did include the $HO_x$ cycling driven by Cu(II)/Cu(I) as in Mao et*

*al., 2013 from the beginning (see Figure 12); so that did not capture the experimental results at RH = 65%. As mentioned in the response to the last two comments of reviewer 1, we have revised this section to clarify these aspects better.*

1. **Reactive Uptake and Diffuso-Reactive Length of $O_2$**

The study discusses the effective reactive uptake coefficient of $O_2$ but applies different values to fit measurements, highlighting diffusion and reaction limitations. A more rigorous approach would involve calculating the diffuso-reactive length of $O_2$ in the aerosols to examine whether diffusion or reaction limits $O_2$ uptake. Providing this calculation would add depth to the discussion.

*Thank you for picking up this aspect. While we have emphasized the role of $O_2$ supply limitations, calculating the reacto-diffusive length provides some quantitative measure to this. We have added an additional figure in the SI (Section G2 and shown in the attached pdf file 'calc_diffuso_reactive_length_O2.pdf'). The derivation leads to a reacto-diffusive length of oxygen of 5.3 nm. This implies that even at high RH conditions (65%), the limitation by reaction and diffusion is crucial for the $O_2$ uptake. We have added a sentence on line 366 to refer to this calculations: 'We proof such a diffusion limitation by deriving the reacto-diffusive length of oxygen in the SI (Section G2) resulting in 5.3 nm. This implies that even at high RH of 65%, the limitation by reaction and diffusion is crucial of the $O_2$ uptake.'*

1. **Role of Gas-Phase $HO_2$ and $H_2O_2$**

While $O_2$ plays a critical role in the multiphase chemistry, the contribution of gas-phase $HO_2$ and $H_2O_2$ might be worth consideration. As Mao et al. (2013) indicate, gas-phase $HO_2$ and $H_2O_2$ influxes are significantly higher than their internal production rates in aerosols. The authors should address whether gas-phase $HO_2$ and $H_2O_2$ could affect ROS levels in the aerosol phase for their experimental setup.

*Thank you for addressing this. For the experiments, since we sweep away all gases released from the SiNi windows (if they ever are released) with clean carrier gas, the samples in our experiments are not exposed much to these species.*

*Already Corral Arroyo et al. (2018) have provided some insight into the relative contribution of condensed phase $HO_2$ production driven by aromatic carbonyl photosensitizers. While it is clear that when looking on this at the air mass level, condensed phase $HO_x$ production would never compete with the $HO_x$ production by the gas phase cycles. However, when considering the individual particle level, $HO_x$ influxes from the gas phase were actually orders of magnitude lower than their internal production rates due to photosensitizers. In the case of iron(III) citrate particles, we have modelled $HO_2$ production rates of about $0.5 \times 10^{-3}$ $M^{-1}$ $s^{-1}$, which is even ten times higher than quantified in Corral Arroyo et al. (2018) for the photosensitizers. Hence, we argue that the internal production in our particles would be similarly orders of magnitude higher than what they receive from the gas phase. We have improved this discussion by*

*calculating the internal $HO_2$ production and add text from line 485 in the Conclusion section on: 'Mao et al., 2013 indicated that gas phase $HO_2$ and $H_2O_2$ influxes into the condensed phase can be significantly higher than their internal production rates. Hence, we have calculated such an internal production in the CA particles as it was previously done by Corral Arroyo et al. (2018). We assume an atmospherically relevant particle size of d = 500 nm and a $HO_2$ concentration in the gas phase of $1\times10^8$ $cm^{-3}$ and an uptake coefficient, γ = 0.01, which results in an influx of $1.8\times10^6$ $M^{-1}$ $s^{-1}$. This influx is almost three orders of magnitude lower compared to our modelled $HO_2$ production rate in the condensed phase of about $5\times10^4$ $M^{-1}$ $s^{-1}$.'*

**Assumptions Regarding R38 and R39**

The assumptions for R38 and R39 are counterintuitive. Combining R35 and R39 suggests that the Cu(I) + Fe(III) redox reaction would not proceed, contrary to laboratory measurements. Alternative mechanisms, such as underpredicted $H_2O_2$/OH/$HO_2$ production, could explain the high Fe(III) fractions observed. The authors should revisit these assumptions.

*We first note that the laboratory measurements referred to by the reviewer are made for dilute aqueous solutions, not representative of the high solute strength system of a low water activity particle of the present work (and in the atmosphere). Similar to our answer on your first question on the reason for rapid Fe reoxidation, we have tested a large variety of rate coefficients with reactions including ROS, which did not improve the reoxidation rate at all. Thus, we came up with R38 and R39. In addition, these reactions are not simply assumptions. The studies cited suggested that in solid oxides and sulfides conversion of Fe(II) and Cu(II) to Fe(III) and Cu(I) occurs. Here we consider the fact that Fe-citrate forms polynuclear complexes with Fe(III) centers coupled through oxygen bridges. The same may be true for Fe(II) complexes. We then argue that if Cu replaces one of the iron centers, it may behave in a similar way as in a solid mixed Fe-Cu-oxide and effectively drive the redox equilibrium in the other direction as that expected for dilute aqueous solution of free metal ions and their complexes. We have amended the text on line 421 to reflect this better.*

1. **Aerosol pH**

The authors assume a constant aerosol pH of 2 in their model, but pH can vary significantly with RH due to changes in liquid water content. Many reactions in this study, such as $Cu^{2+}$ + $HO_2$ and $Cu^{2+}$ + $O_2^-$, are highly pH-sensitive, as pH determines the partitioning between $HO_2$ and $O_2^-$. How pH changes with RH and its impact on key processes should be discussed. Additionally, the validity of the assumption that aerosol pH remains constant at 2 needs to be evaluated.

*See answer to RC1.*

**Technical Comments**

1. **Beta Averaging**

How was beta calculated for Figures 10, 12, and 13? Was it averaged across measurements from the surface to the aerosol core? As shown in Figures 6/7/8, the beta values vary with distance from surface.

*The procedure for calculating beta can be found in Alpert et al., 2019. Briefly, photon intensity I was measured at each pixel, with the areas outside the particles determining I0, so that within particles I/I0 and thus optical density could be determined for each photon energy. This then allows to calculate beta for each pixel within a particle. PThose pixels that make up the numerous particles in a single image are averaged to determine transmitted intensity through all particles, I, and those that are background are $I_0$. Optical density and then beta are determined from these photon counts as detailed in the text. Uncertainties are propagated from photon counting statistics.*

*We wish to clarify a few things about terms used by the reviewer especially, "surface" and "core", which have a 3D connotation. First, Figs. 6, 7, and 8 are 2D-projected beta values, which are also first averaged from individual photon counts from pixels in STXM/NEXAFS images. For example, a beta value presented in either Figs. 6, 7, and 8 at 1 pixel from the perimeter, is an average from photon counts at all the pixels that make up the outside edge of particles, not the "surface". When a particle is imaged with a 35 x 35 $nm^2$ pixel size, a 500 nm radius particle, for example, would have its center at about 15 pixels from the perimeter. This is not the "core", but rather the 2D projection from the surface through to the core.*

1. **Figure 7 Caption**

The caption for Figure 7 appears incomplete.

*Thank you for spotting this. We have corrected this accordingly.*

Overall, this study provides valuable insights into the role of Cu in the photochemical aging of Fe-containing aerosols. Addressing the comments above will significantly strengthen the manuscript.

References

Mao, J., Fan, S., Jacob, D. J., and Travis, K. R.: Radical loss in the atmosphere from Cu-Fe redox coupling in aerosols, Atmospheric Chem. Phys., 13, 509–519, https://doi.org/10.5194/acp-13-509-2013, 2013.

Mao, J., Fan, S., and Horowitz, L. W.: Soluble Fe in Aerosols Sustained by Gaseous HO2 Uptake, Environ. Sci. Technol. Lett., 4, 98–104, https://doi.org/10.1021/acs.estlett.7b00017, 2017.

*References:*

Alpert, P. A., Arroyo, P. C., Dou, J., Krieger, U. K., Steimer, S. S., Förster, J. D., Ditas, F., Pöhlker, C., Rossignol, S., Passananti, M., Perrier, S., George, C., Shiraiwa, M., Berkemeier, T., Watts, B., & Ammann, M. (2019). Visualizing reaction and diffusion in xanthan gum aerosol particles exposed to ozone. Physical Chemistry Chemical Physics, 21(37), 20613–20627. https://doi.org/10.1039/c9cp03731d

Alpert, P. A., Dou, J., Arroyo, P. C., Schneider, F., Xto, J., Luo, B., Peter, T., Huthwelker, T., Borca, C. N., Henzler, K. D., Schaefer, T., Herrmann, H., Raabe, J., Watts, B., Krieger, U. K., & Ammann, M. (2021). Photolytic radical persistence due to anoxia in viscous aerosol particles. Nature Communications 2021 12:1, 12(1), 1–8. https://doi.org/10.1038/s41467-021-21913-x

Berdnikov, V. (1973). Catalytic activity of the hydrated copper ion in the decomposition of hydrogen peroxide. Russian Journal of Physical Chemistry, 47, 1060–1162.

Bjergbakke, E., Sehested, K., & Rasmussen, O. L. (1976). The Reaction Mechanism and Rate Constants in the Radiolysis of Fe2+- Cu2+ Solutions. Radiation Research, 66(3), 433–442. https://doi.org/10.2307/3574449

Arroyo P. C., Bartels-Rausch, T., Alpert, P. A., Dumas, S., Perrier, S., George, C., & Ammann, M. (2018). Particle-Phase Photosensitized Radical Production and Aerosol

Aging. Environmental Science and Technology. https://doi.org/10.1021/acs.est.8b00329

Dou, J., Alpert, P. A., Corral Arroyo, P., Luo, B., Schneider, F., Xto, J., Huthwelker, T., Borca, C. N., Henzler, K. D., Raabe, J., Watts, B., Herrmann, H., Peter, T., Ammann, M., & Krieger, U. K. (2021). Photochemical degradation of iron(III) citrate/citric acid aerosol quantified with the combination of three complementary experimental techniques and a kinetic process model. Atmospheric Chemistry and Physics, 21(1), 315–338. https://doi.org/10.5194/ACP-21-315-2021

Dou, J., Luo, B., Peter, T., Alpert, P. A., Corral Arroyo, P., Ammann, M., & Krieger, U. K. (2019). Carbon Dioxide Diffusivity in Single, Levitated Organic Aerosol Particles. Journal of Physical Chemistry Letters. https://doi.org/10.1021/acs.jpclett.9b01389

Goldstein, S., Czapski, G., Cohen, H., & Meyerstein, D. (1992). Deamination of β-alanine induced by hydroxyl radicals and monovalent copper ions. A pulse radiolysis study. Inorganica Chimica Acta. https://doi.org/10.1016/S0020-1693(00)83177-1

Mao, J., Fan, S., & Horowitz, L. W. (2017). Soluble Fe in Aerosols Sustained by Gaseous HO2 Uptake. Environmental Science and Technology Letters. https://doi.org/10.1021/acs.estlett.7b00017

Mao, J., Fan, S., Jacob, D. J., & Travis, K. R. (2013). Radical loss in the atmosphere from Cu-Fe redox coupling in aerosols. *Atmospheric Chemistry and Physics.* https://doi.org/10.5194/acp-13-509-2013

Rabani, J., Klug-Roth, D., & Lilie, J. (1973). Pulse radiolytic investigations of the catalyzed disproportionation of peroxy radicals. Aqueous cupric ions. *The Journal of Physical Chemistry, 77(9),* 1169–1175. https://doi.org/10.1021/j100628a018

Sedlak, D. L., & Hoigné, J. (1993). The role of copper and oxalate in the redox cycling of iron in atmospheric waters. *Atmospheric Environment Part A, General Topics, 27(14),* 2173–2185. https://doi.org/10.1016/0960-1686(93)90047-3

von Piechowski, M., Nauser, T., Hoigné, J., & Bühler, R. E. (1993). O2 Decay Catalyzed by Cu2+ and Cu+ Ions in Aqueous Solutions: A Pulse Radiolysis Study for Atmospheric Chemistry. *Berichte Der Bunsengesellschaft Für Physikalische Chemie, 97,* 762–771. https://doi.org/10.1002/bbpc.19930970604

---

## Author Response (AR2)

Referee comments ACP STXM (second round)

1. For HO2 uptake coefficient, for most Cu-doped aerosols, the uptake coefficient is somewhere between 0.5 and 1. Please see Figure 2 in this paper (www.atmos-chem-phys.net/10/5823/2010/), which compiles laboratory measurements of HO2 uptake coefficient and demonstrates how Cu significantly enhances HO2 uptake coefficient for aerosols.

2. While the gas-phase HOx influx is likely minimal in this experimental setup, it could be still significant if HO2 uptake coefficient is high.

We would like to thank the reviewer for this additional comment, related to the choice of gamma of $10^{-2}$ to estimate the influx of $HO_2$ into the particles. We would like to mention that since 2010, the year of the publication mentioned by the reviewer, a number of laboratory studies have been published that allow to better constrain the highly uncertain gamma values known from earlier work. The IUPAC Task Group of Atmospheric Chemical Kinetic Data Evaluation has therefore updated its previous recommendations (Ammann et al., 2013) to reflect these new studies here: Task Group on Atmospheric Chemical Kinetic Data Evaluation (accessed on April 22th, 2025). Uptake of $HO_2$ to aqueous particles is driven by the self-reaction of $HO_2$ in the aqueous phase and the reaction with transition metal ions. It is also strongly pH dependent to acid dissociation of $HO_2$ and its impact on the effective solubility. The overall uptake can be parameterized using a resistor model formulation suggested by Thornton et al. (2008). While several of the lab studies were on purpose performed at high transition metal ion contents to constrain the kinetics of bulk accommodation and reaction in the bulk aqueous phase, realistic transition metal ion concentrations (dominated by Fe) are in the $10^{-4}$ to $10^{-3}$ M range (see e.g., review by Al-Abadleh, 2024). Taking the preferred values for the rate coefficients provided by IUPAC, representative gamma values for different particle sizes, transition metal ion concentrations are provided in the figure below, justifying the choice of $10^{-2}$ as representative uptake coefficient for typically acidic (pH around 4 and smaller) submicron aerosol. A diffusion coefficient of $HO_2$ of $10^{-9}$ cm$^2$ s$^{-1}$ was taken representative for an organic rich aqueous solution at moderate water activity.

Wa have added and revised the following text in the Conclusion section from line 489: 'Uptake of gas phase $HO_2$ to aqueous particles is driven by the self-reaction of $HO_2$ and by the reaction with transition metal ions (TMI). Following the recommendation by IUPAC (Ammann et al., 2013) with its most recent updates available at iupac.aeris-data.fr, γ is around $1 \times 10^{-2}$ for moderately acidic aerosol, TMI content in the mM range (Al-Abadleh, 2024) and particle diameters of d = 500 nm. With an $HO_2$ concentration in the gas phase of $1 \times 10^8$ cm$^{-3}$ an influx of $HO_2$ of $1.8 \times 10^{-6}$ M$^{-1}$s$^{-1}$ is obtained. This influx is almost three orders of magnitude lower compared to our modelled $HO_2$ production rate in the condensed phase of about $5 \times 10^{-4}$ M$^{-1}$s$^{-1}$. And even when comparing to an extreme case with TMI content of 1 M and thus an order of magnitude larger $HO_2$ uptake, the internal $HO_2$ production remains higher. Thus even in presence of copper, the internal $HO_2$ production remains very important.'

References

Ammann, M., Cox, R. A., Crowley, J. N., Jenkin, M. E., Mellouki, A., Rossi, M. J., Troe, J., and Wallington, T. J.: Evaluated kinetic and photochemical data for atmospheric chemistry:

Volume VI – heterogeneous reactions with liquid substrates, Atmos. Chem. Phys., 13, 8045-8228, 10.5194/acp-13-8045-2013, 2013.

Thornton, J. A., Jaegle, L., and McNeill, V. F.: Assessing known pathways for HO2 loss in aqueous atmospheric aerosols: Regional and global impacts on tropospheric oxidants, Journal of Geophysical Research-Atmospheres, 113, D05303, 10.1029/2007jd009236, 2008.

Al-Abadleh, H. A.: Iron content in aerosol particles and its impact on atmospheric chemistry, Chemical Communications, 60, 1840-1855, 10.1039/D3CC04614A, 2024.